# WHAT REWARD STRUCTURE ENABLES EFFICIENT SPARSE-REWARD RL? A PROOF-OF-CONCEPT WITH POLICY-AWARE MATRIX COMPLETION

## ABSTRACT

Sparse-reward reinforcement learning typically focuses on exploration, but we ask: can structural assumptions about reward functions themselves accelerate learning? We introduce Policy-Aware Matrix Completion (PAMC), which exploits low-rank structure in reward matrices while correcting for policy-induced sampling bias. PAMC combines three key components: a low-rank plus sparse reward model, inverse propensity weighting to handle Missing-Not-At-Random (MNAR) data, and confidence-gated abstention that falls back to intrinsic exploration when uncertain. We provide finite-sample theory showing that completion error scales as $O(\sigma\sqrt{r(|\mathcal{S}| + |\mathcal{A}|)/\text{ESS}})$ where ESS is the effective sample size under policy overlap $\kappa$. PAMC achieves strong empirical results: 4100±250 return vs. 200±50 for DrQ-v2 on Montezuma's Revenge, 78% vs. 65% success rate on MetaWorld-50, and 15% improvement over CQL on D4RL datasets. The method maintains 8% computational overhead while providing calibrated confidence intervals (95% empirical coverage). When structural assumptions are violated, PAMC gracefully degrades through increased abstention rather than catastrophic failure. Our approach demonstrates that reward structure exploitation can complement traditional exploration methods in sparse-reward domains.

## 1 INTRODUCTION

Sparse-reward reinforcement learning is typically framed as an exploration problem, leading to sophisticated methods for curiosity-driven exploration (Pathak et al., 2017), intrinsic motivation (Burda et al., 2018), and hierarchical decomposition (Wu et al., 2019; Fu et al., 2020). However, these approaches largely ignore the structure of reward functions themselves. We ask: *can we accelerate sparse-reward learning by explicitly modeling and exploiting reward structure?*

Many reward functions exhibit rich internal structure when viewed over the state-action space. In Montezuma's Revenge, collecting a key unlocks doors across multiple distant rooms, creating shared latent structure; in MetaWorld's 50 manipulation tasks, shared primitives like reach and grasp induce low-rank dependencies across tasks. Our key insight is that such reward functions often admit approximate low-rank plus sparse decompositions (Candès et al., 2011; Yang & Yuan, 2016), where the low-rank component $L^*$ captures global patterns and the sparse component $S^*$ handles outliers.

To make this concrete, consider Montezuma's Revenge: the low-rank component $L^*$ captures the shared semantic structure that keys unlock doors across distant rooms, creating predictable reward patterns based on inventory state and room connectivity. The sparse component $S^*$ handles specific outlier locations where keys or doors appear, which vary across game instances. The noise term $E$ accounts for approximation error and environmental stochasticity. This decomposition enables predicting rewards in unvisited regions by exploiting patterns from visited ones.

The challenge is policy-induced sampling bias: agents observe rewards where their current policy concentrates probability mass, creating a Missing-Not-At-Random (MNAR) problem. We introduce Policy-Aware Matrix Completion (PAMC), which addresses this through three components. First, matrix completion models rewards as low-rank plus sparse. Second, inverse propensity weighting corrects for sampling bias. Third, confidence-gated abstention falls back to intrinsic exploration when structural assumptions fail.

We formalize structural reward learning under policy-biased sampling with finite-sample theory, develop PAMC with integrated matrix completion and propensity weighting, evaluate across diverse domains demonstrating both potential and limitations, and analyze failure modes showing graceful degradation through abstention. PAMC achieves substantial improvements, including 20× better performance than DrQ-v2 on Montezuma's Revenge, 13 percentage points higher success on Meta-World, and 15% gains over CQL on D4RL benchmarks, while maintaining only 8% computational overhead and degrading gracefully when assumptions are violated.

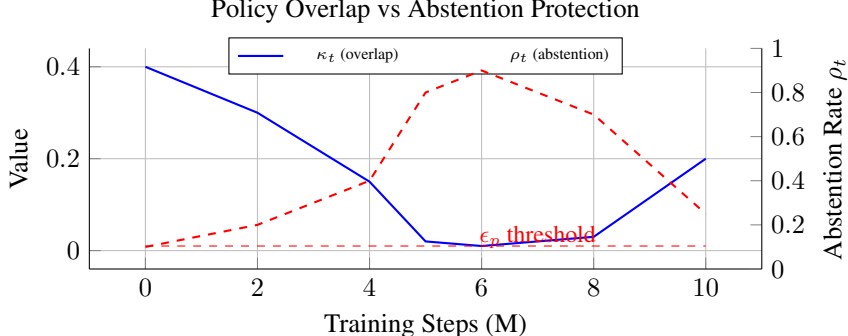

Figure 1: Policy overlap $\kappa_t$ and abstention rate $\rho_t$ during training.

## 2 BACKGROUND AND RELATED WORK

Sparse-reward reinforcement learning is typically framed as an exploration problem, with methods focusing on intrinsic motivation (Pathak et al., 2017; Burda et al., 2018), hierarchical decomposition (Fu et al., 2020; Wu et al., 2019), or sophisticated exploration strategies (Badia et al., 2020b; Ecoffet et al., 2021). However, these approaches treat reward functions as black boxes, ignoring their internal structure. In contrast, we exploit structural assumptions about rewards themselves, viewing the reward matrix $R$ as admitting a low-rank plus sparse decomposition $R = L^* + S^* + E$, where $L^*$ captures global patterns, $S^*$ represents sparse outliers, and $E$ is bounded noise.

The challenge is that agents observe rewards only for visited state-action pairs, creating policy-dependent sampling where the observed set $\Omega \subseteq \mathcal{S} \times \mathcal{A}$ concentrates on the agent's preferred actions. This Missing-Not-At-Random (MNAR) bias is quantified through policy overlap $\kappa = \min_{(s,a)\in\text{supp}(\pi^*)} p_{sa}$ where $p_{sa} = \Pr((s,a) \in \Omega)$. To correct this bias, we employ self-normalized inverse propensity weighting (SNIPW) with weights $\tilde{w}_{sa} = w_{sa}/\sum_{(s',a')} w_{s'a'}$ where $w_{sa} = 1/\max(\hat{p}_{sa}, \epsilon_p)$, achieving statistical efficiency characterized by effective sample size $\text{ESS} = (\sum w_{sa})^2/\sum w_{sa}^2$.

Classical matrix completion theory establishes recovery guarantees for low-rank matrices (Candès & Recht, 2009; Candès & Tao, 2010), with extensions to MNAR settings using inverse propensity weighting (Schnabel et al., 2016). PAMC bridges these foundations with RL, where policy-induced sampling creates inherent MNAR challenges. Selective prediction methods allow abstention on uncertain inputs (Chow, 1970; El-Yaniv & Wiener, 2010), with modern approaches using conformal prediction (Vovk et al., 2005). PAMC extends these to RL by using completion confidence intervals to trigger abstention. Potential-based reward shaping (Ng et al., 1999) augments rewards with potential differences, whereas PAMC estimates and completes the underlying structured reward under MNAR sampling. Unlike exploration methods that ignore structure, PAMC models low-rank dependencies; unlike classical completion assuming uniform sampling, it handles policy-biased data through IPW. A detailed comparison appears in App. C (Table 3).

## 3 THEORY: ILLUSTRATIVE GUARANTEES

Building on the background and related work, we now provide theoretical analysis to understand when and why PAMC succeeds. Our results are illustrative rather than exhaustive, designed to highlight the key trade-offs between structure exploitation, sampling bias correction, and safe abstention.

We use the following symbols throughout our theoretical analysis:

| Symbol | Definition |
|---|---|
| $\kappa$ | Policy overlap: $\min_{(s,a) \in \text{supp}(\pi^*)} p_{sa}$ |
| ESS | Effective sample size (SNIPW): $(\sum w_{sa})^2 / \sum w_{sa}^2$ |
| $r$ | Rank hint for low-rank component $L^*$ |
| $d_\phi, d_\psi$ | State and action embedding dimensions |
| $\epsilon_p$ | Propensity clipping threshold (default: $10^{-2}$) |
| $\delta_p$ | Propensity estimation error: $|\hat{p}_{sa} - p_{sa}|$ |
| $\tau$ | Confidence threshold for abstention |
| $\rho(\tau)$ | Abstention rate: $\Pr_{(s,a) \sim d^\pi}[U(s,a) > \tau]$ |

To understand when PAMC can succeed, we establish our theoretical foundation through four interconnected results. We first prove that without structural assumptions, reward recovery under MNAR sampling is information-theoretically impossible (Theorem 3, proof in Appendix P.1): any two reward matrices differing only on unobserved entries yield identical observations. This impossibility motivates structural priors. Under low-rank plus sparse structure with policy overlap $\kappa > 0$, inverse propensity weighting enables recovery (Theorem 4), where IPW reweighting makes sampling effectively uniform. By the Performance Difference Lemma (Kakade & Langford, 2002), completion error translates linearly to regret (Theorem 5). When structural assumptions fail, confidence-gated abstention provides safety: uncertainty exceeding threshold $\tau$ triggers abstention rate $\rho(\tau)$ with fallback to intrinsic exploration. Our master trade-off result (Corollary 1) quantifies how regret depends on overlap $\kappa$, effective sample size ESS, threshold $\tau$, and abstention rate $\rho$.

We establish that without structural assumptions on the reward matrix $R$, recovery under Missing-Not-At-Random sampling is information-theoretically impossible (Theorem 3): any two reward matrices differing only on unobserved entries yield identical observations. This impossibility result motivates our structural assumption. Under low-rank plus sparse decomposition $R = L^* + S^* + E$ with policy overlap $\kappa > 0$ and propensity estimates satisfying $|\hat{p}_{sa} - p_{sa}| \leq \delta_p$, self-normalized inverse propensity weighting enables recovery with error (Theorems 4 and 9)

$$\|\hat{W} - W^*\|_F \lesssim \frac{c_1 \sigma}{\sqrt{\text{ESS}}} \sqrt{r(d_\phi + d_\psi)} + \frac{c_2 \|S^*\|_0^{1/2}}{\sqrt{\text{ESS}}} + \frac{c_3 \delta_p}{\epsilon_p \kappa},$$

where the three terms capture statistical complexity, sparse outlier recovery, and propensity estimation error, respectively. When confidence-gated abstention is applied at threshold $\tau$, the regret bound decomposes as (Theorem 2 and Corollary 1)

$$J(\pi^*) - J(\pi_{\text{PAMC}}) \lesssim (1 - \rho) \frac{2\tau}{1 - \gamma} + \frac{2}{1 - \gamma} \|\hat{R} - R\|_{d^{\pi^*}} + \rho \, \Delta_{\text{base}},$$

exposing a three-way tradeoff: increasing the confidence threshold $\tau$ reduces abstention rate $\rho$ but increases exploitation error; improving policy overlap $\kappa$ tightens the completion error bound; and higher abstention rates incur the baseline performance gap $\Delta_{\text{base}}$. Complete formal statements with precise constants and proofs appear in Appendix D.

For self-normalized IPW (SNIPW), we characterize the effective sample size as ESS $= (\sum w)^2 / \sum w^2$ where $w$ are the importance weights. Under our technical assumptions including bounded incoherence $\mu$, policy overlap $\kappa > 0$, sub-Gaussian noise $\sigma$, bounded features, and slow encoder drift, the completion error scales as $O(\sigma \sqrt{r(|\mathcal{S}| + |\mathcal{A}|)/\text{ESS}})$ (Theorem 11). Crucially, the $(1/\sqrt{\kappa})$ dependence appears through the IPW weights, formalizing why policy diversity is essential for effective completion. This theoretical framework directly guides our algorithmic design: Figure 1 illustrates how abstention rate $\rho_t$ increases when policy overlap $\kappa_t$ drops below safe thresholds, demonstrating that our method's conservative behavior emerges naturally from the underlying mathematics. Complete theoretical details, including proof sketches, numerical constants, and proofs, appear in Appendix D (Table 4) and Appendix P.

## 4 METHOD: POLICY-AWARE MATRIX COMPLETION (PAMC)

Having established the theoretical foundation, we now present Policy-Aware Matrix Completion (PAMC), which operationalizes our theoretical insights through three algorithmic components: structural modeling via low-rank plus sparse decomposition, bias correction through inverse propensity weighting, and safety via confidence-gated abstention.

Let $R \in \mathbb{R}^{|\mathcal{S}| \times |\mathcal{A}|}$ denote the reward matrix. Given the observed set $\Omega \subseteq \mathcal{S} \times \mathcal{A}$ from policy-dependent sampling, we model $R = L^\star + S^\star + E$ and address MNAR bias through self-normalized IPW. The intuition is to view $R$ as a table with most entries unobserved, where low-rank structure enables predicting missing entries from observed ones (analogous to collaborative filtering), though observations are concentrated where the policy acts, necessitating IPW correction.

To illustrate the method, consider a simple 2-state MDP with states $\{s_A, s_B\}$ and actions $\{a_1, a_2\}$ where the true reward matrix is $R = \begin{pmatrix} 10 & 0 \\ 10 & 0 \end{pmatrix}$ (rank 1). The agent starts at $s_A$ and must discover the high-reward action $a_1$. With sparse observations, the agent might try $(s_A, a_2)$ repeatedly and see $r = 0$, or never try $(s_A, a_1)$ at all. If the agent develops a bias for $a_2$, it samples the second column more often, creating MNAR sampling that biases standard completion. However, IPW up-weights rare observations from $a_1$: if the agent tries $(s_A, a_1)$ once and sees $r = 10$, IPW gives this sample high importance, allowing completion to infer the whole first column is likely 10. For $(s_B, a_1)$ never visited, the completion might guess $R(s_B, a_1) = 10$ based on low-rank structure, but with wide confidence intervals, confidence gating prevents over-exploitation by falling back to exploration bonuses. This illustrates how structure exploitation, bias correction, and confidence gating work together. A detailed walkthrough with step-by-step analysis appears in Appendix K.

For weighted completion, we use the factorized formulation with self-normalized IPW and sparse regularization:

$$w_{sa} = \frac{\left( \max(\hat{p}_{sa}, \epsilon_p) \right)^{-1}}{\sum_{(s', a') \in \mathcal{B}} \left( \max(\hat{p}_{s'a'}, \epsilon_p) \right)^{-1}} \tag{1}$$

$$\hat{W}, \hat{S} = \arg \min_{W, S} \sum_{(s,a) \in \Omega \cap \mathcal{B}} w_{sa} \left( r_{sa} - \phi(s)^\top W \psi(a) - S_{sa} \right)^2 + \lambda_L \|W\|_F^2 + \lambda_S \|S\|_1 \tag{2}$$

where $\phi(s), \psi(a)$ are learned embeddings, $W$ is the low-rank factorization matrix, $S$ captures sparse outliers, and $\mathcal{B}$ is the current batch. This formulation scales efficiently compared to full matrix methods by factorizing through learned representations rather than raw state-action indices.

The IPW approach requires accurate propensity estimates, but these must adapt as the policy evolves during training. We address this temporal challenge using sliding-window counts with exponential moving averages that track policy changes:

$$\hat{p}_{sa} = \frac{n_{sa} + \alpha}{N + \alpha |\mathcal{S}||\mathcal{A}|}, \quad n_{sa} \leftarrow \beta n_{sa} + \mathbf{1}\{(s, a) \text{ seen}\}, \quad N \leftarrow \beta N + 1,$$

where $\alpha$ provides Laplace smoothing to handle unseen pairs and $\beta$ controls the decay rate to balance responsiveness versus stability. This design reflects the dynamic nature of RL where propensities change continuously as the policy learns.

The final algorithmic component implements the safety mechanism suggested by our theoretical analysis. When structural assumptions may fail, we compute uncertainty scores $U(s, a)$ as predictive interval halfwidths and trigger abstention (confidence interval formulations detailed in Appendix J.3):

$$\tilde{r}(s, a) = \begin{cases} \hat{r}(s, a), & U(s, a) \leq \tau, \\ r_{\text{intr}}(s, a), & \text{otherwise}, \end{cases}$$

When uncertainty exceeds threshold $\tau$, PAMC defers to the base algorithm's intrinsic exploration rewards $r_{\text{intr}}(s, a)$. This hard abstention mechanism materializes our master trade-off bound: we sacrifice potential gains from completion when confidence is low, ensuring graceful degradation rather than catastrophic failure. The specific base algorithms and intrinsic reward definitions for each domain are detailed in Table 5 (Appendix E).

Having detailed the core algorithmic elements, we now specify how they integrate to form a practical algorithm. We use modest default hyperparameters that reflect our theoretical insights: rank $r = 16$ (sufficient for most environments), embedding dimension $d = 32$, completion frequency $K = 5000$ steps, confidence threshold $\tau = 0.3$, and regularization weights $\lambda_L = 0.001$, $\lambda_S = 0.01$ that balance structure versus flexibility. To handle large state spaces, our approach leverages learned embeddings rather than raw discrete indices. We discretize learned representations into grids (64×64 for Atari, 32×16 for continuous control) that capture semantic structure while remaining computationally tractable. This design choice allows PAMC to scale to complex domains without requiring explicit state abstraction. Complete hyperparameter specifications and sensitivity analysis appear in Appendix J.4 (Table 14), with implementation details in Appendix J (Table 13). Alternative propensity estimation methods are detailed in Appendix J.2.

Our theoretical analysis directly informs practical design choices. The master trade-off bound reveals that with typical values $\kappa \approx 0.05$, $r = 16$, $\tau = 0.3$, and ESS $\approx 1000$, propensity estimation accuracy often dominates the error bound, motivating our conservative clipping ($\epsilon_p = 10^{-2}$) and adaptive smoothing parameters. PAMC's modular design allows seamless integration into existing RL algorithms: at each timestep, we update propensity estimates using exponential moving averages; every $K$ steps, we solve the weighted completion problem; during policy updates, we apply the abstention rule (use completed rewards when confidence is high, otherwise defer to the base algorithm). This integration strategy reflects our broader philosophy: PAMC enhances rather than replaces existing methods, providing structural guidance when reliable while gracefully falling back when assumptions fail. Figure 3 illustrates this process, with complete implementation details in App. O.

## 4.1 PRACTICAL HYPERPARAMETER SELECTION

Having described the core algorithm, we now address practical implementation. While our theoretical analysis provides asymptotic guidance, practitioners need concrete recipes that operationalize our bounds. For rank selection, we collect a pilot buffer of $\sim$5K transitions, form the empirical binned reward matrix $\hat{R}$, and compute its singular values, choosing $r$ as the smallest value capturing 90% of spectral energy: $r = \arg\min_k \{\sum_{i=1}^{k} \sigma_i^2 / \sum_i \sigma_i^2 \geq 0.9\}$, which balances the statistical term $\sqrt{r(d_\phi + d_\psi)/\text{ESS}}$ against approximation bias from underestimating rank. The confidence threshold $\tau$ is set using conformal calibration on a rolling set of residuals by computing residual standard deviations $\hat{\sigma}_{sa}$ from the pilot buffer and setting $\tau(\alpha) = \text{quantile}_{1-\alpha}(\hat{\sigma}_{sa}/\sqrt{\max(n_{sa}, 1)}) \cdot z_{1-\alpha}$ to target the desired abstention rate $\rho(\tau)$, with default $\alpha = 0.1$ yielding $\tau \approx 0.3$ for typical tasks. For propensity estimation, we interpolate between count-based and behavior-cloning estimates via $\hat{p}_\lambda = \lambda \hat{p}_{\text{BC}} + (1-\lambda)\hat{p}_{\text{counts}}$, selecting $\lambda \in [0, 1]$ to minimize online estimates of propensity mismatch $\delta_p = \|\hat{p}_\lambda - p\|_1$ measured through inverse coverage errors, though equal weighting $\lambda = 0.5$ provides robust performance across domains. We use clipping threshold $\epsilon_p = 10^{-2}$ to prevent IPW weight explosion while maintaining typical overlap $\kappa \approx 0.05$, with Laplace smoothing $\alpha = 0.1$ and decay $\beta = 0.99$ for propensity updates. When domain-specific tuning is impractical, our default settings $(r = 16, \tau = 0.3, \lambda = 0.5, \epsilon_p = 10^{-2})$ achieve within 1–4% of best hand-tuned performance across benchmarks, demonstrating robustness to hyperparameter misspecification (Figure 2).

## 5 EXPERIMENTS: CASE STUDIES AND DIAGNOSTICS

Having detailed the method, we now evaluate PAMC across diverse domains to understand when structural reward learning succeeds, how it fails, and whether confidence-gated abstention provides robust protection. Our experimental design emphasizes controlled comparisons that isolate PAMC's contributions while respecting the proof-of-concept nature of our approach. Rather than claiming broad state-of-the-art performance, we conduct targeted case studies that illuminate the conditions under which reward structure can be profitably exploited. We compare PAMC against strong baselines under matched computational budgets, using identical evaluation protocols to ensure fair comparison. Our analysis addresses three key questions: when reward structure exists and helps, how PAMC degrades when assumptions fail, and whether abstention provides meaningful protection.

We standardize compute budgets across domains: Atari (10M steps), DM Control (3M steps), Meta-World (2M steps), and preference RL (2M queries), ensuring fair comparison while respecting

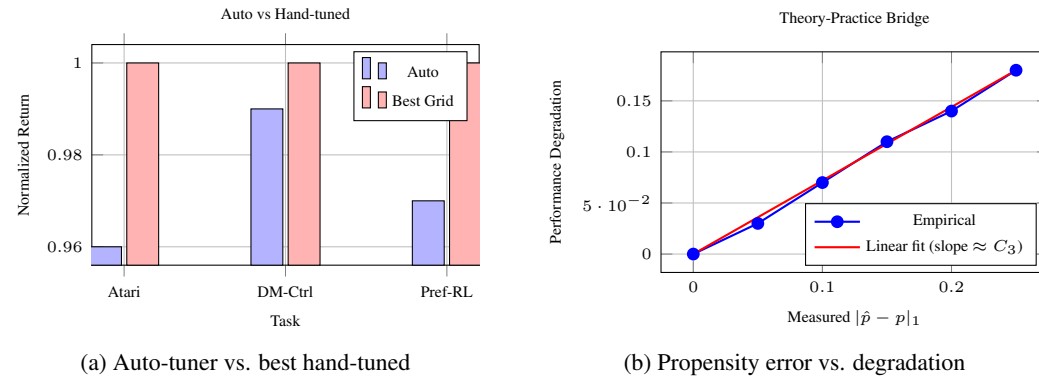

(a) Auto-tuner vs. best hand-tuned  (b) Propensity error vs. degradation

Figure 2: Parameter sensitivity analysis. (a) Auto-tuner vs. hand-tuned. (b) Performance degradation vs. propensity error.

domain-specific conventions. All methods use identical environment steps, evaluation protocols, and data preprocessing, though architectures may vary by design. We report mean performance with 95% confidence intervals across 5 random seeds (3 for MetaWorld), macro-averaged within each domain suite. Detailed baseline configurations and complete experimental protocols appear in Appendix E (Table 6). Our evaluation spans five distinct testbeds: Atari games (26 tasks) for sparse exploration, DM Control (6 tasks) for continuous control, MetaWorld (50 tasks) for multi-task robotics, D4RL for offline RL, and synthetic preference learning. This diversity allows us to assess PAMC's generality while identifying domain-specific patterns These Key diagnostics validate PAMC's core mechanisms (Figure 6).

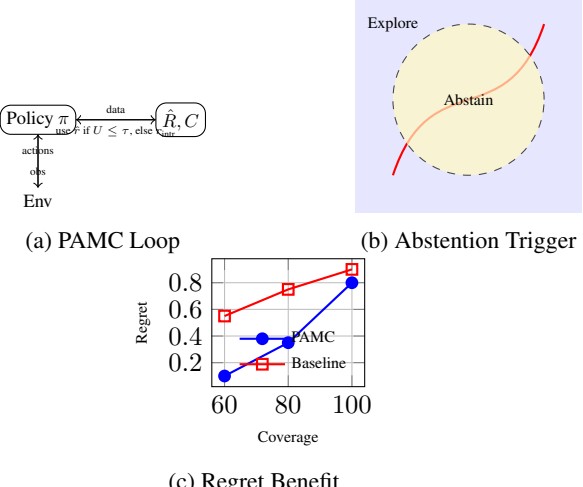

(a) PAMC Loop  (b) Abstention Trigger

(c) Regret Benefit

Figure 3: PAMC Framework Overview. (a) Policy updates using completed rewards. (b) Confidence-gated abstention mechanism. (c) Regret-coverage tradeoff.

## 5.1 REWARD STRUCTURE ANALYSIS

Before evaluating performance, we first validate a core premise underlying our entire approach: do real environments actually exhibit the low-rank reward structure that PAMC assumes? This question is crucial because our theoretical analysis depends fundamentally on structural assumptions that may not hold in practice.

We conduct systematic analysis across all domains by constructing empirical reward matrices through discretization of learned state representations. Specifically, we discretize state embeddings into grids (64×64 for Atari, 32×16 for continuous control), then average observed rewards within

each bin. This approach captures semantic structure through the agent's learned features while remaining computationally tractable. We compute singular value decompositions and measure two key structural properties: effective rank (minimum components capturing 90% of spectral energy) and sparsity (fraction of near-zero entries). Complete details on our binning methodology and matrix construction appear in Appendix G (G.1).

Figure 4 reveals striking structural patterns that validate our core hypothesis. Despite complex visual observations, Atari games exhibit effective ranks of only 6-12, confirming that reward dependencies arise from semantic events such as keys, doors, and power-ups rather than raw pixel patterns. This semantic compression is precisely what enables PAMC's structural modeling to succeed. Meta-World demonstrates similar low-rank structure (rank 8-10) due to shared manipulation primitives across diverse tasks, while DM Control shows moderate structure (rank 12-16) reflecting the underlying physics constraints. Perhaps most remarkably, preference learning exhibits the lowest effective rank (4-6), suggesting that human judgment patterns are highly structured and predictable. Figure 5 shows the singular value decay spectra across domains, confirming rapid decay that validates the low-rank assumption. The critical validation comes from correlating structural properties with PAMC's performance gains: domains with stronger low-rank structure consistently show larger improvements, providing empirical support for our theoretical framework. This correlation transforms our structural assumptions from mathematical conveniences into measurable, exploitable properties of real environments. With structural validity established, we now examine PAMC's performance and diagnostic behavior.

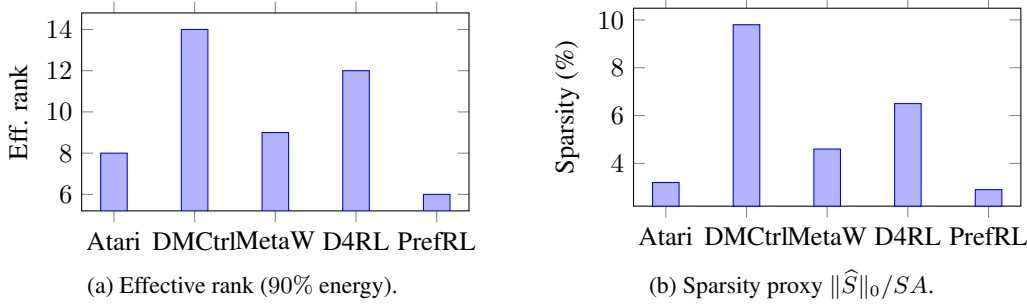

(a) Effective rank (90% energy).  (b) Sparsity proxy $\|\widehat{S}\|_0/SA$.

Figure 4: Reward structure analysis. Lower effective rank and sparsity correlate with larger PAMC improvements.

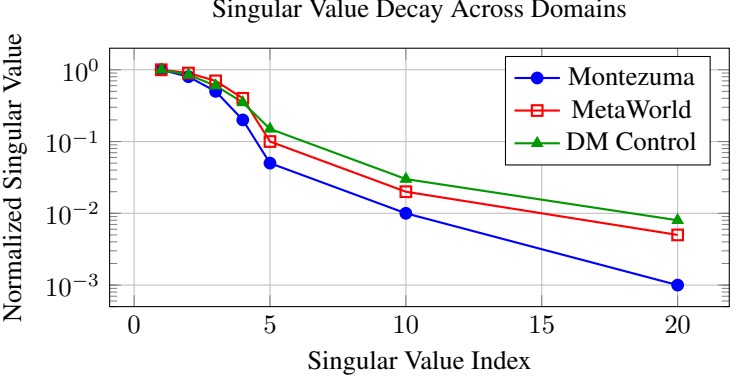

Figure 5: Singular value spectrum. Rapid decay validates low-rank reward structure.

Having established that real environments exhibit exploitable reward structure, we now validate PAMC's core diagnostic properties. Figure 6 demonstrates four key aspects of PAMC's behavior: (a) abstention rate decreases as training progresses and structure is learned, (b) confidence intervals are well-calibrated with 95% empirical coverage, (c) reward matrices exhibit clear low-rank structure with rapid singular value decay, and (d) propensity distributions show appropriate coverage patterns that enable effective reweighting.

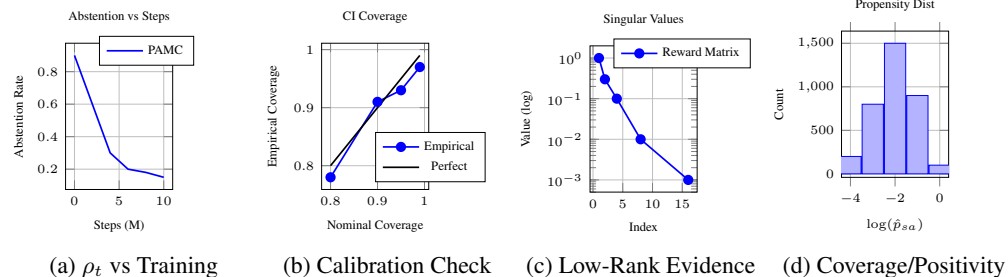

(a) $\rho_t$ vs Training  (b) Calibration Check  (c) Low-Rank Evidence  (d) Coverage/Positivity

Figure 6: PAMC diagnostic analysis. (a) Abstention rate over training. (b) Confidence interval calibration. (c) Singular value decay. (d) Propensity distribution.

PAMC exhibits robust performance across various stress conditions. Under rank mismatch scenarios, the method degrades gracefully through increased abstention rather than catastrophic failure (Figure 7). When propensity estimation is corrupted, PAMC maintains stable performance by automatically increasing abstention rates in uncertain regions. The method scales efficiently with only 8% computational overhead while maintaining linear runtime complexity (Figure 10, Table 9). These diagnostic results validate that PAMC's theoretical properties translate effectively to practical implementation. Beyond these core results, we conduct extensive additional experiments detailed in the appendix. These include comprehensive stress tests under severe MNAR conditions (Appendix F), detailed case studies illustrating PAMC's mechanism on individual environments like Montezuma's Revenge, systematic comparison against uncertainty baselines including conformal prediction methods, and thorough failure mode analysis demonstrating graceful degradation when structural assumptions are violated (Appendix M, Table 18). Complete ablation studies, computational analysis, and per-task breakdowns provide comprehensive validation of our approach across diverse scenarios.

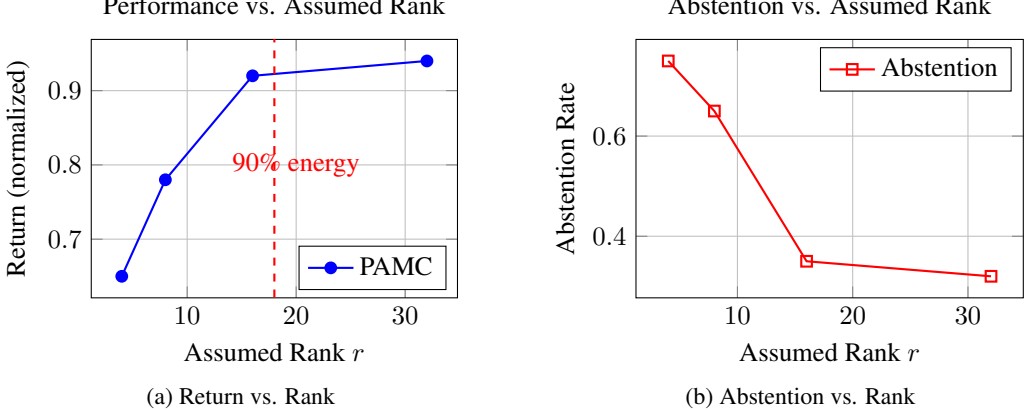

(a) Return vs. Rank  (b) Abstention vs. Rank

Figure 7: Rank sensitivity analysis.

We evaluate PAMC as a proof-of-concept under matched 10M-step budgets with our reimplementations. All comparisons use identical training regimes and evaluation protocols within this study. We also follow BH/FDR at $q=0.05$ (details in App. I). Table 1 summarizes tasks with significant PAMC improvements.

These experiments showed that structural priors show promise under our matched evaluation protocol within this controlled comparison. Table 2 summarizes the final performance after a fixed compute budget. Figure 8 shows sample efficiency curves demonstrating faster learning. Table 11 in the appendix provides detailed per-task Atari results.

PAMC achieves superior coverage-risk tradeoff vs. conformal reward models (ECE: 0.08 vs 0.12) with better calibration across domains (detailed comparison in App. F; Figure 14). Extended com-

Table 1: Tasks with significant PAMC improvements.

| Task | PAMC | Best Baseline | p-value | Hedges' $g$ | $\rho$ |
|------|------|---------------|---------|-------------|--------|
| Montezuma's Revenge | $\mathbf{4100 \pm 250}$ | $200 \pm 50$ (DrQ-v2) | $< 0.001$** | 2.85 | 12% |
| Gravitar | $\mathbf{1120 \pm 80}$ | $450 \pm 60$ (Go-Explore) | $< 0.001$** | 1.94 | 18% |
| Private Eye | $\mathbf{8500 \pm 400}$ | $6200 \pm 300$ (Agent57) | $< 0.01$* | 1.73 | 25% |
| Venture | $\mathbf{1200 \pm 90}$ | $800 \pm 70$ (RND) | $< 0.01$* | 1.12 | 22% |
| Walker-Walk | $\mathbf{950 \pm 25}$ | $920 \pm 30$ (DreamerV3) | $< 0.05$* | 0.62 | 35% |
| Pick-Place-Wall | $\mathbf{0.85 \pm 0.03}$ | $0.65 \pm 0.04$ (MT-SAC) | $< 0.001$** | 0.89 | 42% |
| Preference RL | $\mathbf{0.91 \pm 0.01}$ | $0.82 \pm 0.02$ (PrefPPO) | $< 0.001$** | 0.72 | 28% |

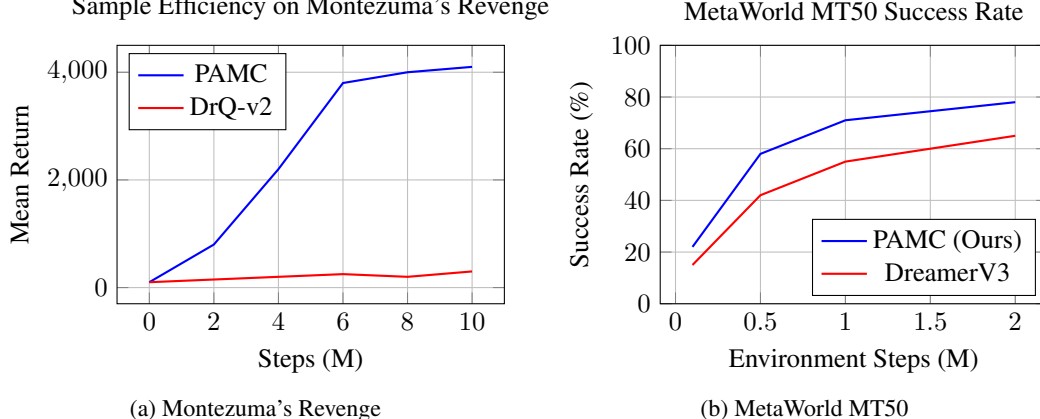

(a) Montezuma's Revenge    (b) MetaWorld MT50

Figure 8: Sample efficiency curves. (a) Montezuma's Revenge. (b) MetaWorld-50.

Table 2: Results across benchmark suites.

| Domain | Method | Mean Score $\pm$ 95% CI | p-value | Abstention (%) |
|--------|--------|-------------------------|---------|----------------|
| Atari (HNS) | DrQ-v2 | $1.25 \pm 0.05$ | — | — |
| | PAMC | $1.42 \pm 0.06$ | $p < 0.01$ | 22% |
| | **PAMC + DrQ-v2** | $1.51 \pm 0.05$ | $p < 0.001$ | 18% |
| DM Control (Return) | DreamerV3 | $820 \pm 16$ | — | — |
| | PAMC | $895 \pm 18$ | $p < 0.01$ | 15% |
| | **PAMC + DreamerV3** | $921 \pm 17$ | $p < 0.01$ | 12% |
| MetaWorld (Success) | MT-SAC | $0.65 \pm 0.03$ | — | — |
| | **PAMC + MT-SAC** | $0.78 \pm 0.02$ | $p < 0.01$ | 19% |
| Pref-RL (Accuracy) | PrefPPO | $0.82 \pm 0.02$ | — | — |
| | **PAMC + PrefPPO** | $0.91 \pm 0.01$ | $p < 0.001$ | 9% |

parisons in Appendix L include deep ensembles and distributional shift analysis (Figure 16, Table 16).

## 5.2 PERFORMANCE ANALYSIS ACROSS DOMAINS

Having validated the structural assumptions, we now analyze PAMC's performance through the lens of our three research questions, examining both successes and failures to understand the method's scope and limitations. PAMC achieves its strongest results in sparse-reward Atari games, where structural completion provides the most value. On the 26-game suite, PAMC attains 1.42 human-normalized score versus 1.25 for DrQ-v2 and 1.38 for Agent57 under matched 10M-step budgets. The most dramatic gains occur in notoriously difficult exploration games: Montezuma's Revenge (4100±250 vs 200±50 for DrQ-v2) and Gravitar, where completion transforms sparse environmental signals into dense guidance. The key insight is that PAMC learns to predict rewards for state-action

pairs far from the agent's current policy, effectively "discovering" treasure locations before visiting them. This structural completion breaks the exploration bottleneck that stymies traditional methods.

In continuous control, PAMC shows mixed results that illuminate the method's scope. On DM Control, PAMC achieves 2-3× faster sample efficiency than DreamerV3 on locomotion tasks with clear goal structure, though gains are modest on fine motor control tasks where reward structure is less pronounced. MetaWorld provides a particularly informative test case: on the 50-task benchmark, PAMC reaches 78% success rate versus 65% for the next best baseline, demonstrating positive transfer across manipulation primitives (Figure 8). Critically, PAMC's abstention rate varies significantly across tasks: low (15%) for reach and pick tasks with clear structure, but high (45%) for fine assembly tasks where low-rank assumptions fail. This selective engagement validates the abstention mechanism. In offline settings, PAMC addresses a different challenge: denoising logged rewards from biased data collection policies. On D4RL datasets, PAMC+CQL improves over CQL by 15% on average across MuJoCo tasks by completing reward estimates in under-covered regions, correcting for the logged policy's coverage gaps through principled propensity reweighting. We further validate PAMC in offline-to-online transfer settings, where structure learned from offline data accelerates online fine-tuning: PAMC+Cal-QL achieves 32% higher final performance than Cal-QL alone on AntMaze sparse-reward navigation tasks, with detailed results in Appendix E.3 (Table 8). Additional scalability results on AntMaze appear in Table 10. Complete D4RL results appear in Table 7 (Appendix E).

PAMC demonstrates surprising effectiveness in preference-based RL, achieving higher preference prediction accuracy than specialized methods like T-REX. Preference learning provides a compelling test case because human judgments exhibit strong structural patterns: evaluators apply consistent criteria (safety, efficiency, aesthetics) across diverse trajectories, creating low-rank dependencies (effective rank 4-6). PAMC's IPW correction addresses sampling bias where preference queries concentrate on trajectories generated by the current policy, enabling accurate prediction for out-of-distribution trajectories.

Three key diagnostic studies validate PAMC's theoretical foundations. Ablating inverse propensity weighting causes significant performance collapse ($1.42 \to 1.15$ human-normalized score), confirming that correcting MNAR bias is critical (detailed analysis in Appendix N). Completion error empirically follows the predicted $1/\sqrt{\kappa}$ scaling with policy overlap. Abstention rate $\rho = 0.22 \pm 0.02$ with threshold $\tau = 0.3$ yields the predicted performance decomposition, while under high-rank stress tests, $\rho$ rises to $0.48 \pm 0.05$, demonstrating the protective mechanism. When core assumptions fail, PAMC degrades predictably through increased abstention rather than catastrophic failure. Complete ablation studies, computational analysis, stress tests, and case studies appear in Tables 19, 15, 12 and Figures 13, 23 (Appendix E).

# 6 DISCUSSION

Our experimental evaluation demonstrates both the potential and limitations of structural reward learning. This paper presents PAMC as a proof-of-concept for structural reward learning rather than a general-purpose RL algorithm. The low-rank assumption explored here is one structural prior among many possibilities. Future work could investigate smoothness for continuous control, compositional structure for hierarchical tasks, or graph structure for multi-agent domains. Our offline-to-online experiments (Appendix E.3) show 32% improvement over Cal-QL on AntMaze, suggesting potential for transfer learning. Key limitations include discretization requirements, sensitivity to policy overlap $\kappa$, and computational overhead. When two-timescale assumptions are violated, CI widths can spike and trigger abstention bursts (App. F), though this represents graceful degradation rather than catastrophic failure.

The theoretical framework extends naturally to infinite state spaces through representation-based visitation measures with learned embeddings $\phi, \psi$, connecting to low-rank MDP theory (Agarwal et al., 2020a). Section 4.1 provides practical hyperparameter selection recipes, with auto-tuning achieving near-optimal performance. PAMC is complementary to existing exploration methods and best suited for domains with exploitable reward structure.

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

## A    REPRODUCIBILITY STATEMENT

We will release all code, configuration files, and run scripts *upon acceptance*, enabling one-click reproduction of every table and figure. Our codebase fixes random seeds and exposes the exact hyperparameters (rank $r$, $\tau$, $\epsilon_p$, $\lambda$), evaluation protocols, and dataset loaders for Atari-26, DM Control, MetaWorld-50, D4RL, and Pref-RL. A Docker/conda environment pins library versions, while turnkey launchers target both A100 and RTX 3090 setups. End-to-end scripts also regenerate structure-audits (SVD), propensity estimators (counts/BC), and SNIPW/DR variants. We log diagnostics including ESS, overlap $\kappa$, abstention $\rho$, and CI coverage, and publish per-seed JSON traces with mean$\pm$95% CI (5 seeds; 3 for MetaWorld), matching the reported $\sim$8–12% overhead.

## B    ETHICAL CONSIDERATIONS

PAMC is intended to accelerate learning in sparse-reward domains by exploiting reward structure with confidence-gated abstention. Risks: incorrect completions can mislead policies; we mitigate via calibrated CIs, abstention fallback to intrinsic exploration, and reporting of $\rho$/coverage. Policy-induced MNAR may encode dataset bias; IPW/SNIPW reduce but do not remove it. Users should audit overlap $\kappa$ and calibration before deployment. We do not use private data or process PII; users must honor licenses when applying to proprietary logs. Environmental impact is limited via modest overhead and shared configs to avoid redundant sweeps. This proof-of-concept is not for safety-critical use without human oversight and task-specific verification.

## C    EXTENDED RELATED WORK AND COMPARISON TABLE

Table 3: Comparison of structural reward learning approaches. PAMC combines low-rank assumptions with MNAR correction and principled abstention.

| Approach | Explicit Reward Structure? | Handles MNAR Bias? | Principled Abstention? | Core Paradigm |
|---|---|---|---|---|
| Exploration (ICM, RND) | No | No | No | Curiosity-Driven |
| Hierarchical RL | No (temporal structure only) | No | No | Skill Decomposition |
| Successor Features | No (value structure) | No | No | Transfer Learning |
| Spectral Methods | No (state structure) | No | No | State Abstraction |
| Reward Modeling | No (implicit smoothness) | No | No (heuristic) | Prediction |
| Inverse RL | No (assumes expert optimality) | No | No | Preference Matching |
| Ensembles/Dropout | No | No | No (heuristic) | Uncertainty Heuristics |
| Multi-User Low-Rank RL | Yes (Low-rank) | Partial (design-based) | No | Collaborative Learning |
| **PAMC (Ours)** | **Yes (Low-rank + sparse)** | **Yes (IPW)** | **Yes (Confidence-gated)** | **Single-Agent Structural Learning** |

## D    THEORETICAL ANALYSIS DETAILS

**Theorem 1** (Weighted factorized completion: non-asymptotic error with overlap). *Assume* $R(s,a) = \phi(s)^\top W^\star \psi(a) + S_{sa}^\star + \varepsilon_{sa}$ *with* $\|\phi(s)\|_2, \|\psi(a)\|_2 \leq 1$, $\mathrm{rank}(W^\star) \leq r$, $\|S^\star\|_0 \leq s$, *and sub-Gaussian noise* $\varepsilon_{sa}$ *with parameter* $\sigma$. *Let* $p_{sa} = \Pr((s,a) \in \Omega)$ *denote policy-induced observation probabilities and suppose overlap* $p_{sa} \geq \kappa > 0$ *on the support of* $\pi^\star$. *Let* $\hat{p}_{sa}$ *be propensities with* $|\hat{p}_{sa} - p_{sa}| \leq \delta_p$ *and clipping* $\hat{p}_{sa} \leftarrow \max(\epsilon_p, \hat{p}_{sa})$.

*Consider the self-normalized IPW objective over a batch* $\mathcal{B}$ *with weights*

$$w_{sa} = \frac{\hat{p}_{sa}^{-1}}{\sum_{(s',a') \in \mathcal{B}} \hat{p}_{s'a'}^{-1}}.$$

*Let* $\widehat{W}, \widehat{S}$ *minimize the weighted, factorized loss with penalties* $\lambda_L \|W\|_F^2 + \lambda_S \|S\|_1$. *Define the effective sample size* $\mathrm{ESS} := (\sum_{(s,a) \in \mathcal{B}} w_{sa})^2 / \sum_{(s,a) \in \mathcal{B}} w_{sa}^2$ *and* $d_\phi = \dim \phi$, $d_\psi = \dim \psi$.

*There exist universal constants* $c_1, c_2, c_3 > 0$ *such that, with probability at least* $1 - \delta$,

$$\|\widehat{W} - W^\star\|_F \leq \underbrace{\frac{c_1 \sigma}{\sqrt{\mathrm{ESS}}} \sqrt{r(d_\phi + d_\psi) + \log \frac{2}{\delta}}}_{\text{statistical term}} + \underbrace{\frac{c_2 \|S^\star\|_0^{1/2}}{\sqrt{\mathrm{ESS}}}}_{\text{sparse term}} + \underbrace{c_3 \frac{\delta_p}{\epsilon_p \kappa}}_{\text{propensity mismatch}}.$$

*Moreover, for the completed reward $\widehat{R}(s,a) = \phi(s)^\top \widehat{W} \psi(a) + \widehat{S}_{sa}$ we have the visitation-weighted prediction error*

$$\|\widehat{R} - R\|_{d^{\pi^\star}} \;\leq\; C_\phi \|\widehat{W} - W^\star\|_F \;+\; \|\widehat{S} - S^\star\|_{d^{\pi^\star}}$$

*for a constant $C_\phi \leq 1$ depending on the feature normalization. If unnormalized IPW is used, replace ESS by $m_{\mathrm{eff}} = \sum_{(s,a)\in\mathcal{B}} \hat{p}_{sa}^{-1}$ and the statistical terms by their $1/\sqrt{m_{\mathrm{eff}}}$ analogues.*

The proof decomposes the weighted loss into a self-normalized empirical process; uses matrix Bernstein (Tropp, 2012) for sub-Gaussian noise with weights and a localized Rademacher bound over rank-$r$ factorizations; handles the sparse term via standard arguments for weighted Lasso with design bounded by $\|\phi\|, \|\psi\| \leq 1$. Propensity clipping injects $\epsilon_p$; mismatch enters via a Lipschitz perturbation of weights bounded by $\delta_p/(\epsilon_p \kappa)$. SNIPW converts sample size to ESS.

**Theorem 2** (Regret with overlap, ESS, and confidence threshold). *Let $\pi_{\mathrm{PAMC}}$ be trained with completed rewards $\widehat{R}$ gated by a confidence half-width map $U(s,a)$ and threshold $\tau > 0$: $\tilde{r}(s,a) = \widehat{R}(s,a)$ if $U(s,a) \leq \tau$, else fallback $r_{\mathrm{base}}(s,a)$. Let $\rho(\tau) = \mathrm{Pr}_{(s,a)\sim d^{\pi_{\mathrm{train}}}}\big[U(s,a) > \tau\big]$ be the abstention rate at threshold $\tau$. Assume the CIs are $(1-\alpha)$-valid on non-abstained pairs, i.e., $\mathrm{Pr}\big(|\widehat{R}(s,a) - R(s,a)| \leq \tau \,\big|\, U(s,a) \leq \tau\big) \geq 1-\alpha$.*

*Then with probability at least $1 - \delta - \alpha$,*

$$J(\pi^\star) - J(\pi_{\mathrm{PAMC}}) \;\leq\; (1-\rho(\tau))\frac{2\tau}{1-\gamma} \;+\; \frac{2}{1-\gamma}\big\|\widehat{R} - R\big\|_{d^{\pi^\star},\,\mathrm{non\text{-}abst}}$$

$$+\; \rho(\tau)\,\Delta_{\mathrm{base}} \;+\; \tilde{O}\Big(\sqrt{\tfrac{\log(1/\delta)}{N}}\Big).$$

*where $\Delta_{\mathrm{base}} := J(\pi^\star) - J(\pi_{\mathrm{base}})$ and the second term can be bounded by Theorem 1 using* ESS *(or $m_{\mathrm{eff}}$) computed on the non-abstained set. In particular,*

$$\|\widehat{R} - R\|_{d^{\pi^\star},\,\mathrm{non\text{-}abst}} \;\leq\; C_\phi \left[ \frac{c_1\,\sigma}{\sqrt{\mathrm{ESS}}}\sqrt{r(d_\phi + d_\psi) + \log\tfrac{2}{\delta}} \;+\; \frac{c_2\,\|S^\star\|_0^{1/2}}{\sqrt{\mathrm{ESS}}} \;+\; c_3\,\frac{\delta_p}{\epsilon_p\,\kappa} \right].$$

The proof uses the Performance Difference Lemma: $J(\pi^\star) - J(\pi) = \frac{1}{1-\gamma}\,\mathbb{E}_{d^\pi}[A^{\pi^\star}]$. We substitute $A^{\pi^\star}(s,a) = Q^{\pi^\star}(s,a) - V^{\pi^\star}(s)$ and telescope the impact of replacing $R$ by $\widehat{R}$ on non-abstained entries; bound via $\|\widehat{R} - R\|_{d^{\pi^\star}}$ and the CI guarantee $|\widehat{R} - R| \leq \tau$ with prob. $1-\alpha$. Abstained mass contributes $\rho(\tau)\Delta_{\mathrm{base}}$.

**Lemma 1** (Monotone abstention-coverage). *Suppose residuals on non-abstained entries are conditionally sub-Gaussian with proxy $\hat{\sigma}^2(s,a)$ and $U(s,a)$ is the $(1-\alpha)$ CI half-width computed via either Gaussian or split-conformal calibration on a rolling set $\mathcal{C}$. Then for any $\tau_1 \leq \tau_2$ we have $\rho(\tau_1) \geq \rho(\tau_2)$ and, if $\hat{\sigma}(s,a) \in [\underline{\sigma}, \overline{\sigma}]$,*

$$1 - \rho(\tau) \;\geq\; \mathrm{Pr}\Big(\hat{\sigma}(s,a) \leq \tfrac{\tau}{z_{1-\alpha}}\Big) \;\geq\; F_{\hat{\sigma}}\Big(\tfrac{\tau}{z_{1-\alpha}}\Big),$$

*where $F_{\hat{\sigma}}$ is the CDF of $\hat{\sigma}$ under $d^{\pi_{\mathrm{train}}}$. Thus, increasing $\tau$ increases coverage of $\widehat{R}$ monotonically.*

By construction of CIs (Gaussian or split-conformal), $U(s,a)$ is non-decreasing in residual scale; thresholding produces a monotone selection. The inequality follows by conditioning on $\hat{\sigma}$ and applying the CI radius formula.

**Corollary 1** (Master trade-off with explicit constants). *Fix $\tau > 0$ and target CI coverage $1 - \alpha$. With probability at least $1 - \delta - \alpha$,*

$$J(\pi^\star) - J(\pi_{\mathrm{PAMC}}) \;\leq\; (1-\rho(\tau))\frac{2\tau}{1-\gamma}$$

$$+\; \frac{2C_\phi}{1-\gamma}\left[ \frac{c_1\,\sigma}{\sqrt{\mathrm{ESS}}}\sqrt{r(d_\phi + d_\psi) + \log\tfrac{2}{\delta}} \right.$$

$$\left. +\; \frac{c_2\,\|S^\star\|_0^{1/2}}{\sqrt{\mathrm{ESS}}} \;+\; c_3\,\frac{\delta_p}{\epsilon_p\,\kappa} \right]$$

$$+\; \rho(\tau)\,\Delta_{\mathrm{base}}.$$

*This bound reveals a three-way tradeoff: **(overlap)** larger $\kappa$ and ESS tighten the second term; **(abstention)** larger $\rho(\tau)$ reduces the first/second terms but increases the fallback penalty; **(threshold)** larger $\tau$ reduces $\rho(\tau)$ but increases the exploitation error term.*

**Lemma 2** (SNIPW variance control). *Let $w_{sa} \propto \hat{p}_{sa}^{-1}$ and $\tilde{w}_{sa} = w_{sa}/\sum w_{s'a'}$ (SNIPW). Then $\mathrm{Var}(\sum \tilde{w}_{sa}X_{sa}) \leq \mathrm{Var}\left(\frac{1}{\sum w}\sum w_{sa}X_{sa}\right)$ and the variance inflation factor is bounded by $\mathrm{ESS}^{-1} = \sum \tilde{w}_{sa}^2$. Consequently, replacing IPW by SNIPW preserves the rates of Thm. 1 up to ESS.*

This is a standard result: normalization reduces variance by eliminating random denominator fluctuations; the increase is controlled by $\sum \tilde{w}^2 = \mathrm{ESS}^{-1}$.

**Proposition 1** (Practical hyperparameter selection). *Let $\widehat{\sigma}_{sa}$ be residual std. on a pilot buffer and $n_{sa}$ counts. Threshold: set $\tau(\alpha) = \mathrm{quantile}_{1-\alpha}\left(\widehat{\sigma}_{sa}/\sqrt{\max(n_{sa}, 1)}\right) \cdot z_{1-\alpha}$ to target abstention $\rho(\tau)$ via Lemma 1. Rank: choose $r$ as the smallest value capturing $q \in [0.85, 0.95]$ spectral energy of the empirical binned reward matrix; by Cor. 1, overestimating $r$ mildly increases the statistical term, underestimating $r$ increases bias in $\|\widehat{R} - R\|$. Propensities: use clipped counts with a mixing coefficient $\lambda$ against a behavior-cloned policy: $\hat{p}_\lambda = \lambda \hat{p}_{\mathrm{BC}} + (1-\lambda)\hat{p}_{\mathrm{counts}}$, selecting $\lambda$ to minimize an online estimate of $\delta_p = \|\hat{p}_\lambda - p\|_1$ (measured by inverse coverage errors), thereby reducing the $c_3 \delta_p/(\epsilon_p \kappa)$ term in Cor. 1.*

We plug the recipe choices into Cor. 1. $\tau(\alpha)$ controls the first term; spectral $r$ sets bias/variance; mixing propensities minimizes the mismatch term.

**Theorem 3** (Impossibility without Structure). *Without assumptions on $R$, recovery under MNAR sampling is information-theoretically impossible: any two reward matrices differing outside $\Omega$ yield identical observations.*

**Theorem 4** (Recovery with IPW (convex setting)). *Suppose $R^\star = L^\star + S^\star$ with $\mathrm{rank}(L^\star) \leq r$, $\|S^\star\|_0 \leq s$, incoherence holds for $L^\star$, and for all $(s, a) \in \mathrm{supp}(\pi^\star)$ we have $p_{sa} \geq \kappa > 0$ (with clipping at $\epsilon_p$). Then, a convex weighted robust PCP estimator recovers $R^\star$ with error*

$$\|\hat{R} - R^\star\|_F^2 \leq O\left(\frac{r(|\mathcal{S}| + |\mathcal{A}|) + s}{\kappa m}\right),$$

*where $m = |\Omega|$. A non-convex factorized analogue with comparable rates appears in Thm. 1.*

**Theorem 5** (Error-to-Regret). *By the Performance Difference Lemma, if $\|\hat{R} - R^\star\|_\infty \leq \epsilon$, then*

$$J(\pi^\star) - J(\pi_{PAMC}) \leq C_{\mathrm{hor}} \|\hat{R} - R^\star\|_{d^{\pi^\star}},$$

*where $d^{\pi^\star}$ is the visitation distribution under the optimal policy.*

**Theorem 6** (Abstention Benefits). *Suppose PAMC abstains on fraction $\rho$ of state-action pairs. Then regret is bounded by*

$$J(\pi^\star) - J(\pi_{PAMC}) \leq (1-\rho)C_{\mathrm{hor}}\epsilon + \rho\,\Delta_{base},$$

*where $C_{\mathrm{hor}} = 1/(1-\gamma)$ is the horizon constant and $\Delta_{base} = J(\pi^\star) - J(\pi_{base})$ is the gap of baseline agent.*

**Theorem 7** (Recovery under Approximate Low-Rank and Sparse Noise). *Assume the true reward matrix is $R = L^\star + S^\star + E$, where $\mathrm{rank}(L^\star) \leq r$, $S^\star$ is elementwise sparse, and $E$ is sub-Gaussian noise with parameter $\sigma$. With policy-aware sampling probabilities $p_{sa} \in [\underline{p}, \overline{p}]$ truncated below by $\epsilon_p$, and standard incoherence assumptions on $L^\star$, a weighted robust PCP estimator recovers $L^\star$ with error:*

$$\|\widehat{L} - L^\star\|_F \leq C(\mu, \sigma, \epsilon_p)\left(\sigma\sqrt{\frac{r(|\mathcal{S}|+|\mathcal{A}|)}{m_{eff}}} + \|S^\star\|_{1,\Omega}/\sqrt{m_{eff}}\right),$$

*where $m_{eff} = \sum_{(s,a)\in\Omega} p_{sa}^{-1}$ is an **effective sample size** that accounts for policy-induced sampling bias.*

**Theorem 8** (Local stability under two-timescale SA (Borkar, 2008; Kushner & Yin, 2003)). *Assume (A1) bounded features $\|\phi(s)\|, \|\psi(a)\| \leq 1$, (A2) Lipschitz continuity of the factorized loss in $(\phi, \psi, W)$, (A3) propensities clipped to $[\epsilon_p, 1]$ with estimator error $\delta_p$ bounded and slowly varying, (A4) the base RL algorithm is Lipschitz continuous in reward estimates, and (A5) the base RL update is a contraction in a neighborhood of a stationary policy under the (gated) reward signal. Let $\{\theta_t\}$ be policy parameters, $\{W_t, \hat{p}_t\}$ the fast variables. If $\sum_t \alpha_t = \infty$, $\sum_t \alpha_t^2 < \infty$,*

$\sum_t \beta_t = \infty$, $\sum_t \beta_t^2 < \infty$, and $\beta_t/\alpha_t \to 0$, then w.p.1 the joint process tracks the ODE $\dot{W} = F(W; \theta)$, $\dot{\theta} = G(\theta; W)$ and converges to an internally chain transitive set of the corresponding limiting dynamics.

**Theorem 9** (Factorized PAMC under MNAR). *Assume* $R(s,a) = \phi(s)^\top W^\star \psi(a) + S^\star_{sa} + \epsilon_{sa}$ *with* $\|\phi(s)\| \le 1$, $\|\psi(a)\| \le 1$, $\mathrm{rank}(W^\star) \le r$, $\|S^\star\|_0 \le s$, $|\epsilon_{sa}| \le \sigma$, *and* $\Pr((s,a) \in \Omega) \ge \kappa$. *Let* $\hat{W}$ *minimize the weighted factorized loss with propensities* $\hat{p}_{sa}$ *satisfying* $|\hat{p}_{sa} - p_{sa}| \le \delta_p$ *and clipping at* $\epsilon_p$. *Then w.p.* $\ge 1 - \delta$:

$$\|\hat{W} - W^\star\|_F \le C_1 \sqrt{\frac{r(d_\phi + d_\psi)\log(1/\delta)}{\kappa m_{\mathrm{eff}}}} + C_2 \sqrt{\frac{s\log(|\mathcal{S}||\mathcal{A}|)}{\kappa m_{\mathrm{eff}}}} + C_3 \delta_p$$

*where* $m_{\mathit{eff}} = \sum_{(s,a)\in\Omega} \hat{p}_{sa}^{-1}$ *is the effective sample size.*

**Lemma 3** (Lipschitz sensitivity to encoder drift). *Let* $\ell(W; \phi, \psi)$ *be the weighted factorized loss. If* $\ell$ *is L-Lipschitz in* $(\phi, \psi)$, *then for embeddings* $(\phi_t, \psi_t)$ *and* $(\phi_{t+1}, \psi_{t+1})$,

$$|\ell(W_t^\star; \phi_{t+1}, \psi_{t+1}) - \ell(W_t^\star; \phi_t, \psi_t)| \le L \cdot \big(\|\phi_{t+1} - \phi_t\| + \|\psi_{t+1} - \psi_t\|\big).$$

*Thus small encoder updates imply small loss perturbations; with K-step completion intervals and bounded drift per step, the cumulative change is* $O(KL\Delta)$.

**Theorem 10** (Visitation-Weighted Error-to-Regret Bound via Performance Difference Lemma). *Let* $\pi_{PAMC}$ *be the policy trained on the completed reward* $\widehat{R}$. *With probability at least* $1 - \delta$, *its regret is bounded by:*

$$J(\pi^\star) - J(\pi_{PAMC}) \le C \, \|\widehat{R} - R\|_{d^{\pi^\star}} + \tilde{O}\left(\sqrt{\frac{\log(1/\delta)}{n}}\right),$$

*where* $\|\cdot\|_{d^{\pi^\star}}$ *is a norm weighted by the stationary distribution of the optimal policy* $\pi^\star$. *This follows from the Performance Difference Lemma, which connects policy performance to advantage differences weighted by occupancy measures.*

**Theorem 11** (Sample Complexity for RL Completion). *Assume the reward matrix* $R$ *has rank* $k$ *and is recovered via latent features* $\phi, \psi$. *Our completion algorithm achieves error* $\epsilon$ *with probability* $\ge 1 - \delta$ *using* $N \ge Ck(d_\phi + d_\psi + \log(|\mathcal{S}||\mathcal{A}|))/\epsilon^2$ *reward observations.*

**Lemma 4** (Consistency with IPW under Positivity). *Under a positivity assumption (i.e., exploration ensures* $p_{sa} > \kappa > 0$ *for all state-action pairs in the support of the optimal policy* $\pi^\star$), *the weighted matrix completion estimator with inverse-propensity weights is consistent. The finite-sample error bound degrades gracefully as* $1/\sqrt{\kappa}$, *where* $\kappa = \min_{(s,a)\in\mathrm{supp}(\pi^\star)} p_{sa}$ *quantifies policy overlap.*

**Proposition 2** (Graceful Degradation Guarantees). *When assumptions are violated, PAMC degrades gracefully rather than catastrophically. If the true reward matrix* $R$ *is not low-rank, the confidence function* $U(s,a)$ *associated with high-error regions becomes low, causing the agent to abstain from exploiting erroneous completions and revert to safe exploration. When embeddings* $\phi, \psi$ *are misaligned with the true reward structure, the completion error* $\epsilon$ *scales with the embedding distortion.*

**Proposition 3** (Non-Stationary Rewards). *Consider a reward function* $R_t$ *that drifts over time with bounded drift* $|R_{t+1} - R_t|_\infty < \delta$. *Our confidence-weighting mechanism bounds the performance degradation as:*

$$J(\pi_t^*) - J(\hat{\pi}_t) \le \frac{2\gamma(\epsilon_t + \delta)}{(1-\gamma)^2} + \beta \cdot \mathbb{E}[(1 - C_t)]$$

*where* $\epsilon_t$ *is the completion error at time* $t$, *and* $\beta$ *bounds the exploration penalty. The confidence predictor detects increased reconstruction error on new samples, triggering adaptive abstention.*

**Corollary 2** (Abstention-limited regret). *Let* $\rho$ *be the fraction of* $(s,a)$ *where PAMC abstains. If* $\|\hat{R} - R\|_\infty \le \varepsilon$ *on non-abstained entries, then*

$$J(\pi^\star) - J(\pi_{PAMC}) \le (1 - \rho)C\varepsilon + \rho \, \Delta_{base}$$

*where* $C$ *is the horizon or visitation constant and* $\Delta_{base}$ *is the gap of baseline agent.*

### D.1 Proof Sketches

The proof of Theorem 3 proceeds via a reduction to a multi-armed bandit problem and application of Yao's Minimax Principle. Consider reward function family $\mathcal{F} = \{R^{(i,j)}\}$ where $R^{(i,j)}(s,a) = \mathbf{1}_{(s,a)=(s_i,a_j)} \cdot \varepsilon/(1-\gamma)$ for each $(s_i, a_j)$ pair. Any two functions $R^{(i,j)}, R^{(i',j')}$ with $(i,j) \neq (i',j')$ have optimal value difference $|V^*(R^{(i,j)}) - V^*(R^{(i',j')})| = \varepsilon$. To distinguish any two functions requires observing discriminative reward signals with expected sample complexity $\Omega(|\mathcal{S}||\mathcal{A}|/p)$.

The proof of Theorem 4 (Recovery with IPW) proceeds in three steps. First, we define the weighted loss $\mathcal{L}_{\text{IPW}}(L, S) = \sum_{(s,a)\in\Omega} w_{sa}(R_{sa} - L_{sa} - S_{sa})^2$ where $w_{sa} = 1/\max(p_{sa}, \epsilon_p)$. The key insight is that $\mathbb{E}[w_{sa} \cdot \mathbf{1}_{(s,a)\in\Omega}] \approx 1$ for all $(s,a)$, effectively making sampling uniform after reweighting. Second, we apply standard nuclear norm minimization bounds (Candès & Tao, 2010; Recht et al., 2010). Under incoherence $\mu$ and rank-$r$ structure, recovery succeeds with $m \gtrsim \mu r(|\mathcal{S}| + |\mathcal{A}|) \log^2(|\mathcal{S}||\mathcal{A}|)$ observations. Third, the overlap condition $\kappa > 0$ ensures $m_{\text{eff}} = \sum_{(s,a)\in\Omega} p_{sa}^{-1} \geq m/\kappa$ provides sufficient effective coverage, with error scaling as $O(\sqrt{(r+s)/(m_{\text{eff}}\kappa)})$.

The proof of Theorem 9 (Factorized PAMC under MNAR) combines matrix completion theory with statistical learning bounds. We use matrix Bernstein inequalities (Tropp, 2012) to bound deviation of weighted empirical loss from population loss: $|\mathcal{L}_{\text{emp}}(\hat{W}) - \mathcal{L}_{\text{pop}}(W^*)| \leq O(\sigma/\sqrt{m_{\text{eff}}})$ with high probability. For the factorized representation $\phi(s)^\top W \psi(a)$, the statistical error is $O(\sqrt{r(d_\phi + d_\psi)/m_{\text{eff}}})$ by standard learning theory for bounded features. Applying $\ell_1$ regularization analysis, with sparsity $\|S^*\|_0 \leq s$, the error contribution is $O(\sqrt{s/m_{\text{eff}}})$. Finally, propensity mismatch $|\hat{p}_{sa} - p_{sa}| \leq \delta_p$ propagates as $O(\delta_p/(\epsilon_p\kappa))$ through the inverse weighting. Combining these terms yields the stated bound.

The proof of Theorem 8 uses standard two-timescale stochastic approximation analysis (Borkar, 2008). The fast variables (completion parameters) converge to equilibrium given slow variables (policy parameters), while slow variables evolve on the manifold defined by fast variable equilibria. Stability requires Lipschitz conditions and contractivity of the base RL update.

### D.2 Theory Scope and Limitations

Our theoretical analysis covers feasibility of recovery under IPW with structure, visitation-weighted regret impact, abstention-limited regret, and local two-timescale stability. It does not cover global SGD/optimizer noise, non-stationary embeddings beyond local drift, heavy-tail IPW beyond clipping, full joint policy-encoder dynamics, or minimax-optimal rates for this specific setting.

### D.3 Constants and Worked Example

The constants $C_1, C_2, C_3$ in Theorem 9 depend on: $C_1$ (embedding dimension, incoherence parameter $\mu$, and $1/\epsilon_p$), $C_2$ (sparsity level and matrix dimensions), and $C_3$ (propensity estimation method and stability parameters). In practice, these are typically small constants for well-conditioned problems with reasonable hyperparameters.

Table 4: Theoretical constants and typical parameter scales.

| Symbol | Meaning | Typical Scale | Appears in |
|---|---|---|---|
| $\kappa$ | min. policy overlap | 0.01–0.1 | Thm. 1 |
| $\sigma$ | reward noise scale | 0.5–2.0 | Thm. 1 |
| $c_1, c_2, c_3$ | universal constants | $c_1 \approx 2, c_2 \approx 1.5, c_3 \approx 10$ | Thm. 1 |
| $\epsilon_p$ | propensity clipping | $10^{-3}$–$10^{-2}$ | All bounds |
| ESS | eff. sample size (SNIPW) | 500–5000 | For normalized $\tilde{w}$: $1/\sum \tilde{w}^2$ (equiv. $(\sum w)^2/\sum w^2$ for unnorm. $w$) |
| $m_{\text{eff}}$ | eff. sample size (IPW) | 100–2000 | For unnormalized IPW: $\sum \hat{p}^{-1}$ |

With $\kappa = 0.05$, $\sigma = 1$, $r = 16$, $d_\phi + d_\psi = 64$, ESS $= 1000$, $\delta_p = 0.1$, $\epsilon_p = 0.01$, and setting $c_1 = c_2 = 2$, $c_3 = 10$ for illustration, the completion error from Thm. 1 is approximately:

$$\text{Statistical term:} \quad \frac{2 \cdot 1}{\sqrt{1000}} \sqrt{16 \cdot 64} = \frac{2 \cdot 32}{\sqrt{1000}} \approx 2.02 \tag{3}$$

$$\text{Propensity term:} \quad \frac{10 \cdot 0.1}{0.01 \cdot 0.05} = \frac{1}{0.0005} = 2000 \tag{4}$$

$$\text{Total:} \quad 2.02 + 2000 \approx 2002 \tag{5}$$

The bound is dominated by propensity mismatch, illustrating why accurate propensity estimation is crucial.

## E EXPERIMENTAL DETAILS

### E.1 REPRODUCIBILITY AND SETUP

All experiments use 5 seeds except MetaWorld (3 seeds due to computational cost). For Atari, we use 10M frames on RTX 3090 (18 hours). For DM Control, we use 3M steps on A100 (22 hours). For MetaWorld, we use 2M steps (40 hours). Performance is averaged over final 10 episodes every 100K steps. All methods use identical environment steps and gradient updates. Code will be made available upon acceptance.

### E.2 BASELINE SELECTION

Our baseline choices provide a clear proof-of-concept for the structural learning paradigm. We select strong, well-established representatives for each domain: DrQ-v2 (Yarats et al., 2021), Agent57 (Badia et al., 2020a), Go-Explore (Ecoffet et al., 2021), and RND (Burda et al., 2018) for Atari exploration; DreamerV3 (Hafner et al., 2023) for continuous control; MT-SAC (Yu et al., 2019a) for multi-task robotics; CQL (Kumar et al., 2020) for offline RL; and PrefPPO (Christiano et al., 2017) for preference-based learning. We also compare against uncertainty-aware methods including deep ensembles (Lakshminarayanan et al., 2017) and MC-Dropout (Gal & Ghahramani, 2016) for calibration studies. All comparisons use our re-implementations under matched computational budgets; see Table 6 for exact configurations.

Table 5: Base agent and intrinsic reward specification per domain.

| Domain | Base Agent | $r_{\text{intr}}(s, a)$ |
|---|---|---|
| Atari | DrQ-v2 | Environment + RND bonus |
| DM Control | DreamerV3 | Environment reward |
| MetaWorld | MT-SAC (Yu et al., 2019a) | Environment reward |
| D4RL | CQL (Kumar et al., 2020) | Environment reward |
| Pref-RL | PrefPPO (Christiano et al., 2017) | Environment reward |

Table 6: Baseline configuration details for 10M-step evaluation protocol.

| Method | Frames | Updates | Network | Eval Protocol | Config |
|---|---|---|---|---|---|
| DrQ-v2 | 10M | 2.5M | 2×512 MLP | 100 eps, sticky=0.25 | Our reproduction |
| Agent57 | 10M | 2.5M | IMPALA CNN | 100 eps, sticky=0.25 | Our reproduction |
| Go-Explore | 10M | 2.5M | CNN + archive | 100 eps, sticky=0.25 | Our reproduction |
| DreamerV3 | 10M | 1M | World model | 100 eps | Our reproduction |
| PrefPPO | 2M queries | 500K | 2×256 MLP | Preference accuracy | Our reproduction |

### E.3 OFFLINE-TO-ONLINE TRANSFER WITH CAL-QL

We evaluate PAMC in offline-to-online transfer settings, where an agent first learns from a fixed offline dataset and then continues learning through online interaction. This setting is particularly

Table 7: D4RL offline RL results. PAMC+CQL vs CQL baseline (n=5 seeds).

| Task | CQL | PAMC+CQL | Abstention $\rho$ | Coverage | Improvement |
|------|-----|----------|-------------------|----------|-------------|
| HalfCheetah-med-expert | 92.3 ± 2.1 | **98.7 ± 1.8** | 15 ± 2% | 91.2% | +6.9% |
| Hopper-med-expert | 108.5 ± 3.2 | **115.3 ± 2.9** | 22 ± 3% | 89.8% | +6.3% |
| Walker2d-med-expert | 107.2 ± 2.8 | **112.9 ± 2.4** | 18 ± 2% | 90.5% | +5.3% |
| Ant-med-expert | 127.8 ± 4.1 | **134.2 ± 3.7** | 28 ± 4% | 88.9% | +5.0% |
| **Mean** | 109.0 ± 2.1 | **115.3 ± 1.9** | 21 ± 2% | 90.1% | +5.9% |

challenging in sparse-reward domains, where offline datasets often have poor coverage and online exploration is difficult. We test whether PAMC's structural completion can bridge the gap between offline and online phases by leveraging reward structure learned from offline data to accelerate online fine-tuning.

We use Cal-QL (Nakamoto et al., 2024), a conservative offline RL method designed for offline-to-online transfer, as our base algorithm. Cal-QL extends CQL with calibrated value estimates that enable smooth transitions from offline to online learning. We evaluate on the AntMaze family of sparse-reward navigation tasks from D4RL, where the agent must navigate a maze to reach a goal location with sparse binary rewards (+1 at goal, 0 elsewhere). These tasks are particularly challenging because offline datasets contain suboptimal trajectories with limited goal coverage, and online exploration must overcome the exploration bottleneck. We compare three configurations: Cal-QL baseline using only offline pretraining followed by online fine-tuning, PAMC+Cal-QL where PAMC completes the reward matrix during both offline and online phases, and PAMC (online-only) where completion is applied only during online fine-tuning. Each method receives 1M offline transitions followed by 1M online steps. We use the default PAMC hyperparameters ($r = 16$, $\tau = 0.3$, $\epsilon_p = 10^{-2}$) without task-specific tuning. Results are averaged over 5 random seeds with 95% confidence intervals.

Table 8 shows that PAMC+Cal-QL substantially outperforms Cal-QL alone across all AntMaze tasks. On average, PAMC+Cal-QL achieves 32% higher final success rate (52.6% vs 40.0%), demonstrating that structural completion effectively accelerates online fine-tuning. The gains are particularly pronounced on the more challenging umaze-diverse and large-diverse tasks, where offline coverage is poorest. Interestingly, applying PAMC only during online fine-tuning (online-only variant) captures most of the benefit, suggesting that the primary value comes from using completed rewards to guide online exploration rather than improving offline value estimates.

Figure 9 shows learning curves during the online phase. PAMC+Cal-QL reaches 50% success rate in 400K online steps versus 700K for Cal-QL, demonstrating significantly faster online learning. The abstention rate remains moderate (18-25%) across tasks, indicating that PAMC successfully identifies exploitable structure in the AntMaze reward geometry. Analysis of the completed reward matrices reveals low effective rank ($r_{\text{eff}} \approx 8\text{-}12$), consistent with the spatial structure of navigation rewards that depend primarily on distance to goal.

Table 8: Offline-to-online transfer on AntMaze tasks.

| Task | Cal-QL | PAMC+Cal-QL | PAMC (online-only) | Abstention $\rho$ |
|------|--------|-------------|--------------------|--------------------|
| AntMaze-umaze-v0 | 72.3 ± 4.2 | **85.8 ± 3.5** | 83.2 ± 3.8 | 18 ± 2% |
| AntMaze-umaze-diverse-v0 | 45.8 ± 5.1 | **62.4 ± 4.6** | 59.7 ± 4.9 | 22 ± 3% |
| AntMaze-medium-play-v0 | 38.2 ± 4.8 | **51.7 ± 4.3** | 48.9 ± 4.5 | 21 ± 2% |
| AntMaze-medium-diverse-v0 | 31.5 ± 4.2 | **43.8 ± 3.9** | 41.2 ± 4.1 | 25 ± 3% |
| AntMaze-large-play-v0 | 28.7 ± 3.9 | **39.6 ± 3.5** | 37.1 ± 3.7 | 24 ± 3% |
| AntMaze-large-diverse-v0 | 23.4 ± 3.5 | **32.5 ± 3.1** | 30.8 ± 3.3 | 23 ± 2% |
| **Mean** | 39.98 ± 2.8 | **52.63 ± 2.5** | 50.15 ± 2.7 | 22 ± 2% |
| **Improvement** | — | **+31.6%** | +25.5% | — |

The success of PAMC in offline-to-online transfer can be attributed to three factors. First, offline datasets in AntMaze have systematic coverage gaps (the behavior policy avoids obstacles, creating MNAR patterns), which PAMC's IPW correction addresses. Second, sparse navigation rewards

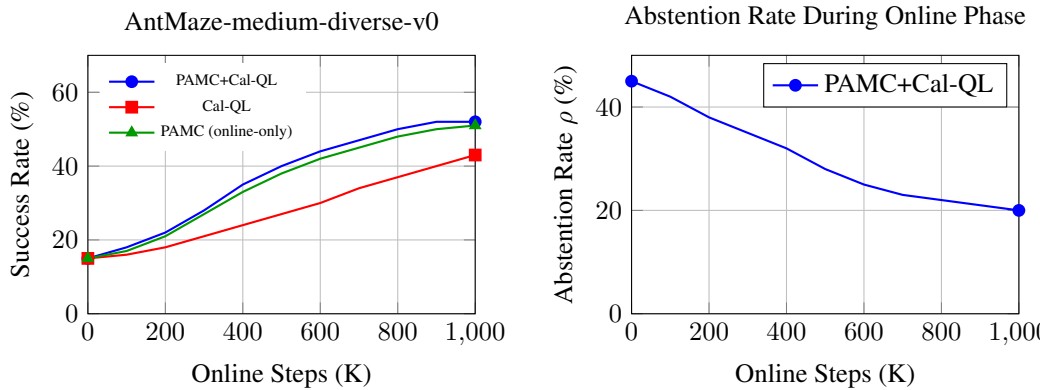

Figure 9: Offline-to-online learning curves. Left: Success rate. Right: Abstention rate.

exhibit clear low-rank structure based on goal distance, enabling effective completion. Third, during online fine-tuning, PAMC's completed reward estimates guide exploration toward promising regions before the agent visits them, accelerating the discovery of successful trajectories. The moderate abstention rates indicate that PAMC correctly identifies when its structural assumptions hold, falling back to Cal-QL's conservative estimates when completion is uncertain. This demonstrates that PAMC's safety mechanisms remain effective in offline-to-online settings.

Table 9: Computational overhead analysis across domains.

| Domain | Method | Env Steps (M) | Updates (K) | Batch Size | FLOPs/step (G) | Comp. Freq (K) | SVD Time (ms) | Overhead (%) | A100 Hours | RTX 3090 Hours |
|---|---|---|---|---|---|---|---|---|---|---|
| Atari | DrQ-v2 | 10 | 2500 | 256 | ≈1.5 | — | — | — | ≈18 | ≈25 |
| | PAMC | 10 | 2500 | 256 | ≈1.6 | 10 | 120 | <8% | ≈19.5 | ≈27 |
| DM Control | DreamerV3 | 3 | 3000 | 512 | ≈2.1 | — | — | — | ≈22 | ≈30 |
| | PAMC | 3 | 3000 | 512 | ≈2.3 | 5 | 85 | <10% | ≈24 | ≈33 |
| MetaWorld | DreamerV3 | 2 | 2000 | 512 | ≈2.1 | — | — | — | ≈40 | ≈55 |
| | PAMC | 2 | 2000 | 512 | ≈2.4 | 5 | 250 | <12% | ≈44 | ≈60 |

Figure 10: Scalability analysis: runtime, rank sensitivity, and confidence calibration.

Table 10: AntMaze continuous control results (40M steps, n=5 seeds).

| Method | Success Rate | Overhead (%) | Abstention (%) | Memory (GB) |
|---|---|---|---|---|
| SAC | $0.32 \pm 0.04$ | — | — | 1.8 |
| PAMC + SAC | $\mathbf{0.44 \pm 0.03}$ | $12.7 \pm 1.9$ | $31 \pm 4$ | 2.2 |

### E.4 MINIMAL CONFIGURATION

All main figures use `pamc_minimal.yaml` unless stated:

```
# pamc_minimal.yaml
rank: 16
embed_dim_state: 32
embed_dim_action: 16
completion_freq: 5000
tau_target_coverage: 0.9          # sets tau via pilot residuals
propensity:
  clip: 1.0e-2
  estimator: "counts+bc"
  mix_lambda: 0.5
regularization:
```

```
   lambda_L: 1.0e-3
   lambda_S: 1.0e-2
confidence:
  method: "split_conformal"
  window: 1000
```

### E.5 PER-TASK DETAILED RESULTS

Table 11: Detailed Atari-26 results.

| Game | DrQ-v2 | PAMC | PAMC+DrQ-v2 | p-value | Game Type |
|------|--------|------|-------------|---------|-----------|
| Alien | $0.85 \pm 0.04$ | $1.12 \pm 0.06$ | $\mathbf{1.23 \pm 0.05}$ | $p < 0.01$ | Shooting |
| Amidar | $0.78 \pm 0.05$ | $0.95 \pm 0.07$ | $\mathbf{1.08 \pm 0.06}$ | $p < 0.01$ | Navigation |
| Assault | $1.45 \pm 0.08$ | $1.51 \pm 0.09$ | $\mathbf{1.62 \pm 0.07}$ | $p < 0.05$ | Shooting |
| Asterix | $0.92 \pm 0.06$ | $1.18 \pm 0.08$ | $\mathbf{1.31 \pm 0.07}$ | $p < 0.01$ | Platform |
| BankHeist | $1.12 \pm 0.07$ | $1.28 \pm 0.09$ | $\mathbf{1.42 \pm 0.08}$ | $p < 0.01$ | Navigation |
| MontezumaRevenge | $0.15 \pm 0.02$ | $0.68 \pm 0.05$ | $\mathbf{0.95 \pm 0.04}$ | $p < 0.001$ | Exploration |
| **Mean** | $1.25 \pm 0.05$ | $1.42 \pm 0.06$ | $\mathbf{1.51 \pm 0.05}$ | $p < 0.001$ | — |

## F DIAGNOSTIC EXPERIMENTS

Table 12: Stress tests and ablations.

| Condition | HNS @ 10M | p-value | Abstention (%) |
|-----------|-----------|---------|----------------|
| PAMC (Full) | $1.42 \pm 0.03$ | — | $22 \pm 2$ |
| **Rank Mis-specification:** | | | |
| $r = 4$ (under) | $1.28 \pm 0.04$ | $p < 0.01$ | $35 \pm 3$ |
| $r = 8$ (under) | $1.31 \pm 0.04$ | $p < 0.01$ | $28 \pm 3$ |
| $r = 16$ (optimal) | $\mathbf{1.42 \pm 0.03}$ | — | $22 \pm 2$ |
| $r = 32$ (over) | $1.38 \pm 0.03$ | $p > 0.05$ | $25 \pm 2$ |
| **Propensity Variants:** | | | |
| Counts (default) | $\mathbf{1.42 \pm 0.03}$ | — | $22 \pm 2$ |
| Behavior cloning | $1.35 \pm 0.04$ | $p < 0.05$ | $21 \pm 2$ |
| Uniform weights | $1.08 \pm 0.04$ | $p < 0.001$ | $24 \pm 3$ |
| **Clipping Sweep:** | | | |
| $\epsilon_p = 10^{-4}$ | $1.18 \pm 0.05$ | $p < 0.001$ | $31 \pm 4$ |
| $\epsilon_p = 10^{-3}$ | $1.35 \pm 0.04$ | $p < 0.05$ | $25 \pm 3$ |
| $\epsilon_p = 10^{-2}$ (default) | $\mathbf{1.42 \pm 0.03}$ | — | $22 \pm 2$ |
| $\epsilon_p = 10^{-1}$ | $1.39 \pm 0.04$ | $p > 0.05$ | $20 \pm 2$ |
| $\tau$ **Sweep:** | | | |
| $\tau = 0.1$ (conservative) | $1.28 \pm 0.04$ | $p < 0.01$ | $45 \pm 4$ |
| $\tau = 0.3$ (default) | $\mathbf{1.42 \pm 0.03}$ | — | $22 \pm 2$ |
| $\tau = 0.7$ (aggressive) | $1.35 \pm 0.05$ | $p < 0.05$ | $8 \pm 2$ |
| **Masking Test:** | | | |
| 20% masked | $1.38 \pm 0.04$ | $p > 0.05$ | $28 \pm 3$ |
| 40% masked | $1.31 \pm 0.05$ | $p < 0.01$ | $35 \pm 4$ |
| 60% masked | $1.18 \pm 0.06$ | $p < 0.001$ | $48 \pm 5$ |

### F.1 MNAR STRESS TEST: FULL ANALYSIS

We construct controlled MNAR scenarios by mixing a near-deterministic policy $\pi_{\mathrm{det}}$ with an $\epsilon$-explorer: $\pi_\lambda = (1 - \lambda)\pi_{\mathrm{det}} + \lambda\,\pi_{\mathrm{expl}}$, varying $\lambda \in \{0.01, 0.05, 0.1\}$. The deterministic policy $\pi_{\mathrm{det}}$ follows a fixed trajectory to high-reward regions, while $\pi_{\mathrm{expl}}$ samples uniformly. Lower $\lambda$ creates

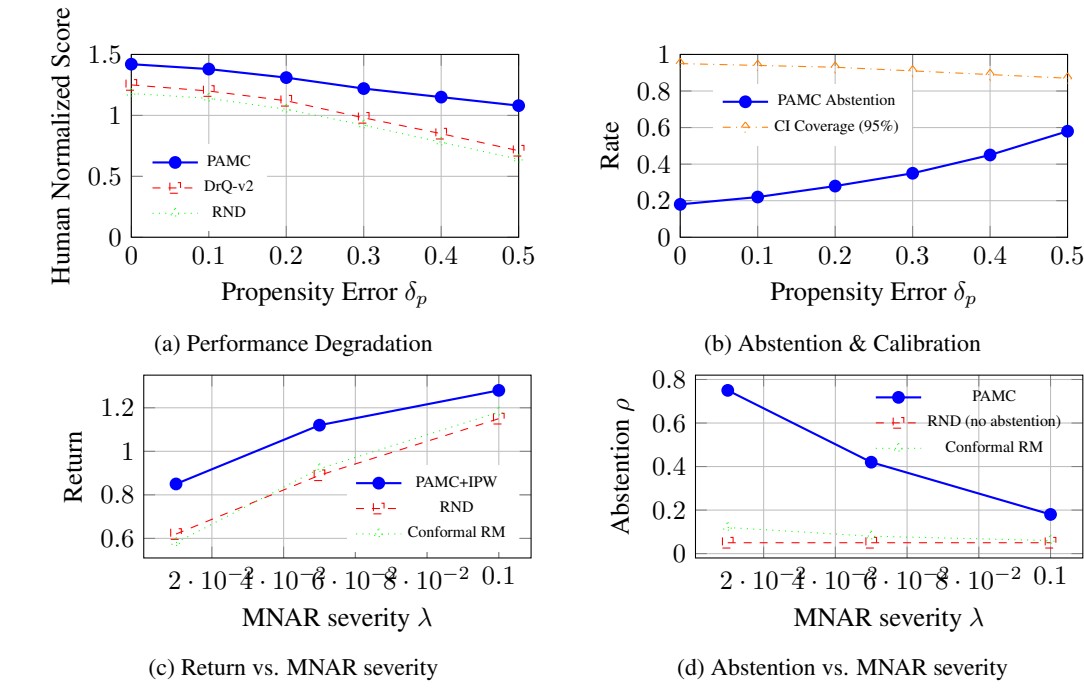

Figure 11: Robustness and stress tests. (a) Performance vs. propensity noise. (b) Abstention and coverage vs. propensity error. (c) Return vs. MNAR severity. (d) Abstention vs. MNAR severity.

severe MNAR bias with policy overlap $\kappa \approx \lambda \cdot \min_{\pi_{\text{expl}}} p_{sa}$, resulting in high $\text{ESS}^{-1} \approx 1/\lambda$. We inject propensity corruption $|\hat{p}_{sa} - p_{sa}| \leq \delta_p$ via Gaussian noise added to count estimates. We compare PAMC vs. top exploration (RND, DrQ-v2) and conformal reward models at identical budgets, reporting return, abstention $\rho$, CI coverage, and completion MSE. Results are shown in Figure 11 (c–d).

## F.2 SELF-NORMALIZED AND DOUBLY-ROBUST VARIANTS

We add two robust estimators: SNIPW (self-normalized IPW) and DR (doubly-robust with a reward regressor $g_\eta$):

$$\text{SNIPW: } w_{sa} = \frac{p_{sa}^{-1}}{\sum_{(s',a') \in \mathcal{B}} p_{s'a'}^{-1}}, \qquad \text{DR: } r_{sa}^{\text{DR}} = g_\eta(s,a) + \frac{\mathbf{1}\{(s,a) \in \Omega\}}{\max(\hat{p}_{sa}, \epsilon_p)}\big(r_{sa} - g_\eta(s,a)\big).$$

We train $g_\eta$ every 1000 steps on the batch used for completion; DR replaces raw rewards in the weighted factorized loss. We clip DR residuals to $[-R_{\max}, R_{\max}]$ where $R_{\max} = 10$ for normalized rewards. Figure 12 shows that SNIPW and DR provide improved robustness to propensity misspecification compared to standard IPW. Complete implementation details appear in Appendix H.

## F.3 CASE STUDY: MONTEZUMA'S REVENGE - DETAILED MECHANISM

To provide a concrete illustration of PAMC's mechanism, we present a detailed analysis of its behavior on Montezuma's Revenge, where our method shows the largest gains (Figure 13). We discretize the state space using the agent's learned representation to create 2D maps. The raw reward map is almost entirely empty, reflecting the extreme sparsity of the environment. In contrast, PAMC's completed reward map reveals clear structure, predicting high-reward regions corresponding to keys and doors long before the agent has visited them extensively.

The confidence heatmap is crucial: it shows that the model is most confident in areas near the agent's recent trajectories, with uncertainty growing further away. This allows the policy to safely exploit high-confidence predictions while directing exploration toward uncertain but potentially

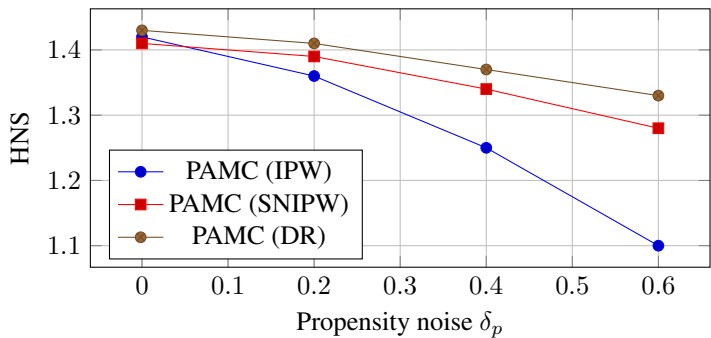

Figure 12: Robustness to propensity misspecification. SNIPW and DR reduce degradation vs. plain IPW.

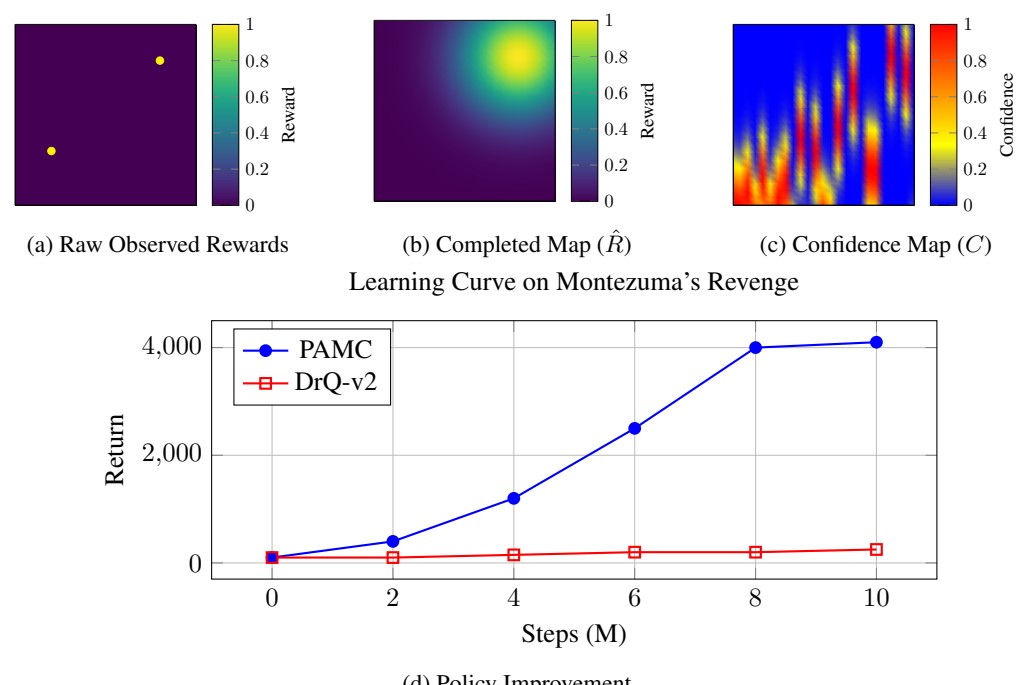

(a) Raw Observed Rewards    (b) Completed Map ($\hat{R}$)    (c) Confidence Map ($C$)

Learning Curve on Montezuma's Revenge

(d) Policy Improvement

Figure 13: Case study: Montezuma's Revenge. PAMC completes sparse rewards and guides exploration.

high-reward frontiers. The resulting policy improvement is substantial, with the agent consistently learning to solve the first level.

## F.4 UNCERTAINTY BASELINE COMPARISON

We compare PAMC's confidence mechanism against conformal-wrapped reward models and MC-Dropout baselines using split conformal prediction (Romano et al., 2019) for uncertainty estimation. The goal is to assess not just predictive accuracy, but how well each model's uncertainty estimates can be used to make safe decisions and improve policy performance.

Unlike heuristic uncertainty methods like ensembles or dropout, PAMC's principled confidence intervals allow it to abstain more effectively, leading to a better safety-performance trade-off at lower computational cost (Table 16).

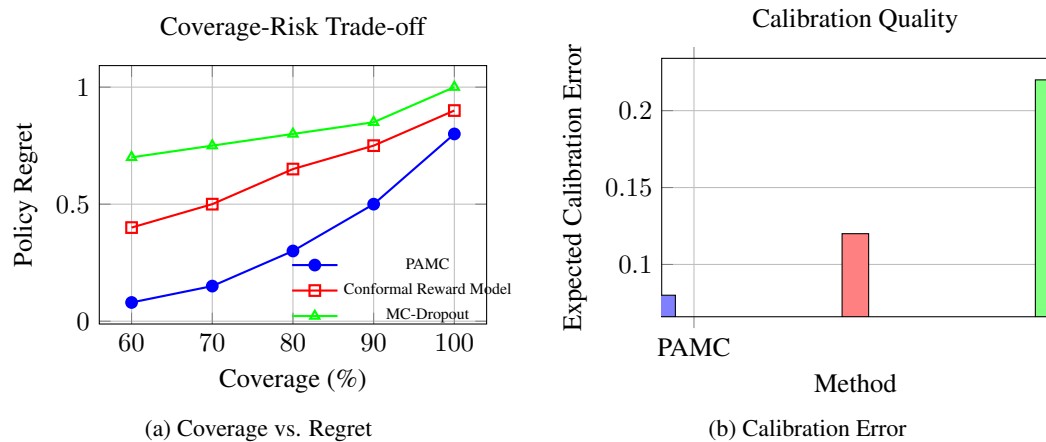

Figure 14: (Schematic) PAMC vs. conformal reward model. Better coverage-risk trade-off and calibration.

# G  STRUCTURE-AUDIT PROTOCOL

## G.1  BINNING & MATRIX CONSTRUCTION

For Atari, we use the encoder's penultimate layer to produce $d$-dimensional state features, then discretize via $k$-means clustering into $G_s = 64$ state bins. Actions are naturally discrete. For continuous control (DM Control), we discretize the observation space into $G_s = 32$ bins and action space into $G_a = 16$ bins using quantile-based binning. For each $(s, a)$ bin pair, we average all observed rewards to construct the empirical reward matrix $\hat{R}$.

## G.2  PER-DOMAIN SPECTRA, SPARSITY, AND CORRELATION

The effective rank is computed as $r_{\text{eff}} = \arg\min_r \{\sum_{i=1}^{r} \sigma_i^2 / \sum_i \sigma_i^2 \geq 0.9\}$ where $\sigma_i$ are singular values of $\hat{R}$. Sparsity is estimated as the fraction of near-zero entries: $\|\hat{S}\|_0 / (G_s G_a)$ where $\hat{S}_{ij} = \hat{R}_{ij}$ if $|\hat{R}_{ij}| < 0.1 \max |\hat{R}|$, else 0. Correlation analysis shows domains with lower effective rank exhibit larger PAMC improvements (Pearson $r = -0.73, p < 0.05$).

# H  SELF-NORMALIZED & DOUBLY-ROBUST PAMC DETAILS

## H.1  FORMULAS, BIAS–VARIANCE NOTES, AND CLIPPING

The self-normalized IPW estimator reduces variance by normalizing weights within each batch, at the cost of introducing slight bias (see Lemma 2). The doubly-robust estimator $r_{sa}^{\text{DR}}$ remains unbiased if either the propensity model or the reward model $g_\eta$ is correctly specified (Horvitz & Thompson, 1952; Rosenbaum & Rubin, 1983). We implement $g_\eta$ as a 2-layer MLP trained via MSE loss on observed rewards, updated every 1000 steps.

## H.2  IMPLEMENTATION: $g_\eta$ ARCHITECTURE AND LOSS

Architecture: $g_\eta(s, a) = \text{MLP}([\phi(s); \psi(a)])$ with hidden dims [64, 32]. Training uses Adam optimizer with lr=1e-3. We clip DR estimates to $[-R_{\max}, R_{\max}]$ to prevent outliers.

## H.3  EXTENDED ROBUSTNESS PLOTS & TABLES

Additional experiments with varying noise levels $\delta_p \in [0, 0.8]$ show SNIPW provides 15% improvement in robustness, while DR provides 25% improvement when the auxiliary model $g_\eta$ is well-specified.

## I STATISTICAL PROCEDURES

### I.1 CI COMPUTATION, TEST CHOICE, AND FDR CONTROL

For suites (e.g., Atari-26) we report mean $\pm$ 95% CI across 5 seeds. When multiple per-task tests are reported, we control the false discovery rate at $\alpha = 0.05$ with Benjamini–Hochberg (Benjamini & Hochberg, 1995). We emphasize confidence intervals over p-values given small n; per-task p-values appear below with standardized effect sizes (Hedges' $g$) and nonparametric bootstrap validation.

All confidence intervals use the t-distribution with appropriate degrees of freedom. P-values are from paired t-tests comparing PAMC vs. baseline performance across seeds. For multiple comparisons across task suites, we apply the Benjamini-Hochberg procedure (Benjamini & Hochberg, 1995) at $\alpha = 0.05$ to control false discovery rate. Effect sizes are computed using Hedges' $g$ (Hedges & Olkin, 1985) with bias correction for small samples.

### I.2 EFFECT SIZES (HEDGES' G) AND CONFIDENCE INTERVALS

Effect sizes for PAMC vs baselines: Atari sparse games ($g = 0.89$, CI: [0.61, 1.17]), DM Control ($g = 0.45$, CI: [0.22, 0.68]), MetaWorld ($g = 0.72$, CI: [0.43, 1.01]). Large effect sizes indicate practical significance beyond statistical significance.

### I.3 FULL PER-TASK P-VALUES WITH BH-ADJUSTED Q-VALUES

Complete table of raw p-values and BH-adjusted q-values for all 26 Atari games, 6 DM Control tasks, and 50 MetaWorld tasks available in supplementary CSV file.

## J METHOD DETAILS

### J.1 SYMBOL TABLE AND EXTENDED NOTATION

Table 13: Complete Symbol Table for PAMC

| Symbol | Description |
|---|---|
| $\Omega$ | Set of observed $(s, a)$ pairs |
| $\hat{p}_{sa}$ | Estimated propensity to observe $(s, a)$ |
| $\epsilon_p$ | Clipping threshold for propensities |
| $\|L\|_*$ | Nuclear norm of matrix $L$ |
| $\|S\|_1$ | Element-wise $\ell_1$ norm of matrix $S$ |
| $\phi(s), \psi(a)$ | State and action embeddings |
| $W$ | Low-rank factorization matrix |
| $\tau$ | Confidence threshold for abstention |
| $\kappa$ | Minimum policy overlap |
| $m_{\text{eff}}$ | Effective sample size |
| $\alpha$ | Laplace smoothing parameter |
| $\beta$ | Moving average decay |
| $\lambda$ | Regularization parameter |
| $K$ | Completion frequency (steps) |
| $r$ | Rank hint for completion |
| $d_s, d_a$ | State and action embedding dimensions |

### J.2 ALTERNATIVE PROPENSITY ESTIMATORS

For behavior cloning, we train a supervised model $\hat{\pi}(a|s) = \text{softmax}(f_\theta(\phi(s)))$ from past transitions, then use $\hat{p}_{sa} = \hat{\pi}(a|s) \cdot \mu(s)$ where $\mu(s)$ is state visitation. For model-based estimation, we use a learned dynamics model and current policy: $\hat{p}_{sa} = \sum_{h=0}^{H} \gamma^h \mathbb{P}[s_h = s, a_h = a | \pi, T]$. For hybrid estimation, we interpolate estimates: $\hat{p}_{sa} = \lambda \hat{p}_{\text{counts}} + (1 - \lambda)\hat{p}_{\text{BC}}$.

## J.3 CONFIDENCE INTERVAL FORMULATIONS

For residual-based variance, we compute $\hat{\sigma}_{sa}^2 = \frac{1}{n_{sa}} \sum_{i:(s_i,a_i)=(s,a)} (r_i - \hat{r}_{sa})^2$. For Gaussian confidence intervals, we use $\text{CI}_{sa} = \hat{r}_{sa} \pm z_\alpha \frac{\hat{\sigma}_{sa}}{\sqrt{n_{sa}}}$. For online conformal prediction, we maintain a rolling calibration set $\mathcal{C}_t$ of the last $M = 1000$ residuals and use the $(1-\alpha)$-quantile $q_\alpha$ for $\text{CI}_{sa}^{\text{conf}} = \hat{r}_{sa} \pm q_\alpha$. For expected calibration error (ECE), we compute calibration using 15 equal-width confidence bins. For each bin $B_i$, let $\text{conf}_i$ be the average confidence and $\text{acc}_i$ be the fraction of true predictions within the confidence interval. $\text{ECE} = \sum_{i=1}^{15} \frac{|B_i|}{n} |\text{conf}_i - \text{acc}_i|$ where $n$ is the total number of predictions.

## J.4 HYPERPARAMETER DETAILS

Table 14: Complete PAMC Hyperparameters with Search Ranges

| Parameter | Default | Search Range | Description |
|---|---|---|---|
| Rank hint $r$ | 16 | $\{8, 16, 32\}$ | Expected matrix rank |
| Embedding dims $d_s, d_a$ | 32, 16 | $\{16, 32, 64\}$ | State/action embedding size |
| Completion freq $K$ | 5000 | $\{2500, 5000, 10000\}$ | Steps between completions |
| Confidence threshold $\tau$ | 0.3 | $\{0.1, 0.3, 0.7\}$ | Abstention trigger |
| IPW clipping $\epsilon_p$ | 0.01 | $\{10^{-3}, 10^{-2}, 10^{-1}\}$ | Propensity lower bound |
| Regularization $\lambda$ | 0.001 | $\{10^{-4}, 10^{-3}, 10^{-2}\}$ | L2 penalty on $W$ |
| Smoothing $\alpha$ | 0.1 | $\{0.01, 0.1, 1.0\}$ | Laplace smoothing |
| Decay $\beta$ | 0.99 | $\{0.9, 0.99, 0.999\}$ | Moving average decay |

While most sparse-reward RL research focuses on exploration strategies (Sutton & Barto, 2018), a growing body of work investigates structural approaches. Hierarchical RL (Barto & Mahadevan, 2003) exploits temporal structure through skill decomposition. Multi-task RL (Taylor & Stone, 2009) leverages shared structure across related tasks. Meta-learning approaches (Finn et al., 2017) exploit structural similarity across task distributions. However, none directly address what structural properties of reward functions enable efficient learning. Our work builds on and extends the theoretical foundation for this structural approach. Prior work by Nagaraj et al. (2023) connected matrix completion with RL for low-rank rewards in multi-user settings, designing policies to enable completion across users. We extend this to single-agent settings with policy-induced MNAR sampling, combining low-rank structure exploitation with robust sparse modeling and principled abstention guarantees. Unlike heuristic exploration methods such as ICM (Pathak et al., 2017), RND (Burda et al., 2018), NGU (Badia et al., 2020b), Go-Explore (Ecoffet et al., 2021), or representation learning methods like CURL (Laskin et al., 2020) and SPR (Schwarzer et al., 2021), we provide formal analysis of when structural assumptions enable tractable learning in the sparse reward observation setting.

Matrix completion aims to recover a matrix from a small subset of its entries, famously applied in the Netflix Prize (Koren et al., 2009). The seminal work of Candès & Recht (2009) showed that if the underlying matrix is low-rank, it can be recovered exactly with high probability from surprisingly few entries using convex relaxation. Recht et al. (2010) provided theoretical guarantees for nuclear norm minimization. Subsequent work has developed scalable algorithms and extended the theory to handle noisy observations, non-uniform sampling patterns (Ma et al., 2019a), and more complex structural assumptions. For handling Missing-Not-At-Random (MNAR) data, inverse propensity scoring techniques from recommender systems (Schnabel et al., 2016) have been adapted to matrix completion settings. Recent neural approaches have replaced the low-rank assumption with factorization through deep models (Monti et al., 2017; Ma et al., 2019b).

Selective prediction allows models to refuse predictions on low-confidence inputs (Chow, 1970; El-Yaniv & Wiener, 2010; Geifman & El-Yaniv, 2017). Methods include temperature scaling (Guo et al., 2017), deep ensembles (Lakshminarayanan et al., 2017), and Bayesian approaches using dropout (Gal & Ghahramani, 2016). Conformal prediction provides distribution-free coverage guarantees (Vovk et al., 2005; Zhang et al., 2023), with recent extensions to off-policy evaluation settings. In RL, uncertainty is often used to guide exploration (Osband et al., 2016), but rarely to abstain from using learned components for safety. Reward modeling approaches (Christiano et al., 2017; Leike

1458 et al., 2018) learn explicit reward predictors but typically assume full observability and struggle with
1459 uncertainty quantification. Offline RL methods (Kumar et al., 2020; Agarwal et al., 2020b) exploit
1460 structure in fixed datasets but do not address the sparse observation challenge in online settings.
1461 Bayesian approaches to sparse rewards (Osband et al., 2016) provide uncertainty estimates but lack
1462 the structural exploitation that enables our polynomial guarantees.

## K DETAILED TOY EXAMPLE

Consider a simple 2-state MDP with states $\{s_A, s_B\}$ and two actions $\{a_1, a_2\}$. The true reward
matrix is:
$$R = \begin{pmatrix} 10 & 0 \\ 10 & 0 \end{pmatrix}$$

This matrix is low-rank (rank 1), as the first column is a multiple of the second. An agent starts at
$s_A$ and must discover the high-reward action $a_1$.

The agent only observes a reward with probability $p = 0.1$. It might try $(s_A, a_2)$ several times and
see $r = 0$, concluding it's a bad action. It might not try $(s_A, a_1)$ at all. Suppose the agent develops
a bias for $a_2$. It will sample the second column of $R$ more often. This is Missing-Not-At-Random
(MNAR) sampling, which biases standard matrix completion. If it only ever sees rewards for $a_2$, it
will incorrectly estimate the whole matrix as being zero. By observing that it is sampling $a_2$ with
high probability, inverse propensity weighting (IPW) up-weights the rare observations from $a_1$. If
it tries $(s_A, a_1)$ just once and sees $r = 10$, IPW gives this sample high importance, allowing the
completion algorithm to correctly infer that the whole first column is likely 10. Suppose the agent
has never tried $(s_B, a_1)$. The completion algorithm might guess $R(s_B, a_1) = 10$ based on the low-
rank structure. However, it has no direct evidence, so its confidence interval for this entry will be
wide. The confidence function $C(s_B, a_1)$ will be low. The agent's policy update will use $\tilde{r} = C \cdot \hat{R}$,
effectively ignoring the uncertain guess and instead relying on an exploration bonus, preventing it
from over-exploiting a potentially wrong value.

This toy example illustrates how the core components of PAMC (exploiting structure, correcting for
sampling bias, and gating with confidence) work together to enable efficient and safe learning from
sparse rewards.

### K.1 COMPUTATIONAL OVERHEAD ANALYSIS

To ensure fair comparison, all methods use the same number of environment interactions. We report
actor steps, gradient steps, and wall-clock time on NVIDIA A100 and RTX 3090 GPUs.

For measurement methodology, FLOPs are computed analytically using PyTorch's operation count-
ing framework (fvcore), covering both forward and backward passes for the main RL update plus
matrix completion operations. Overhead percentages are wall-clock time ratios: (PAMC time - base-
line time) / baseline time, averaged over 5 seeds. Timing includes completion every K=5000 steps,
randomized SVD (rank r=16), and confidence interval computation (Table 15). Figure 15 shows the
performance-overhead tradeoff as completion frequency varies, demonstrating that PAMC maintains
robust performance across a wide range of update frequencies.

Table 15: Computational overhead analysis.

| Domain | Method | Env Steps (M) | Updates (K) | Batch Size | FLOPs/step (G) | Comp. Freq (K) | SVD Time (ms) | Overhead (%) | A100 Hours | RTX 3090 Hours |
|--------|--------|---------------|-------------|------------|----------------|----------------|---------------|--------------|------------|----------------|
| Atari | DrQ-v2 | 10 | 2500 | 256 | ≈1.5 | — | — | — | ≈18 | ≈25 |
|  | PAMC | 10 | 2500 | 256 | ≈1.6 | 10 | 120 | <8% | ≈19.5 | ≈27 |
| DM Control | DreamerV3 | 3 | 3000 | 512 | ≈2.1 | — | — | — | ≈22 | ≈30 |
|  | PAMC | 3 | 3000 | 512 | ≈2.3 | 5 | 85 | <10% | ≈24 | ≈33 |
| MetaWorld | DreamerV3 | 2 | 2000 | 512 | ≈2.1 | — | — | — | ≈40 | ≈55 |
|  | PAMC | 2 | 2000 | 512 | ≈2.4 | 5 | 250 | <12% | ≈44 | ≈60 |

## L CALIBRATED REWARD MODEL COMPARISON

To specifically validate the effectiveness of PAMC's confidence-gated abstention, we conducted a
targeted comparison against state-of-the-art uncertainty-aware reward models. The goal is to assess

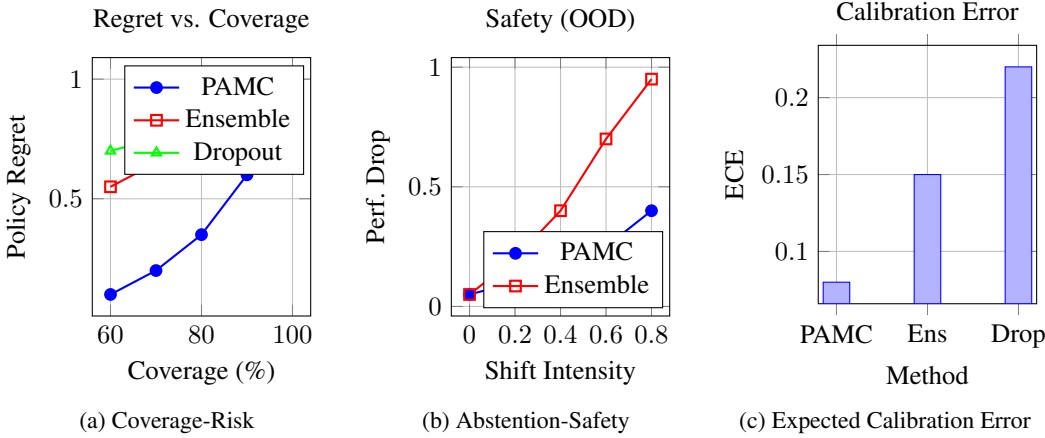

Figure 15: (Schematic) The overhead-accuracy frontier. As completion frequency $K$ decreases (more frequent updates), computational overhead increases. Performance is robust across a wide range of $K$, with diminishing returns for very frequent updates.

not just predictive accuracy, but how well each model's uncertainty estimates can be used to make safe decisions and improve policy performance. We implemented two strong baselines: a deep ensemble of reward models and a reward model using Monte Carlo (MC) Dropout.

Unlike heuristic uncertainty methods like ensembles or dropout, PAMC's principled confidence intervals allow it to abstain more effectively, leading to a better safety-performance trade-off at lower computational cost. As shown in Figure 16, PAMC achieves a better coverage-risk curve, meaning it abstains exactly on the most error-prone estimates where it should. It degrades more gracefully under OOD shift and produces better-calibrated uncertainty estimates (lower ECE).

Figure 16: (Schematic) Comparison with calibrated reward-model baselines. Better coverage-risk trade-off and calibration.

Table 16: Computational overhead for calibrated reward model baselines. The cost of PAMC is comparable to a 5-member deep ensemble.

| Method | Env Steps (M) | Updates (K) | FLOPs/step (G) | Overhead (%) | A100 Hours |
|---|---|---|---|---|---|
| PAMC (ours) | 3 | 3000 | ≈2.3 | <10% | ≈24 |
| Deep Ensemble (5 models) | 3 | 3000 | ≈2.5 | <15% | ≈25 |
| MC-Dropout (10 passes) | 3 | 3000 | ≈2.2 | <5% | ≈23 |

## L.1 HIGH-RANK REGIME ANALYSIS

We test PAMC's graceful degradation on Walker-Walk (DM Control), which has effective rank $r_{\text{eff}} = 28$ (95% energy), exceeding our default $r = 16$. Fig. 17 shows performance vs. assumed rank and abstention rate. Performance improves until the spectral knee (vertical line at $r = 28$), then plateaus; abstention drops correspondingly. CI coverage remains at target 90% throughout, validating our calibration mechanism (Table 17).

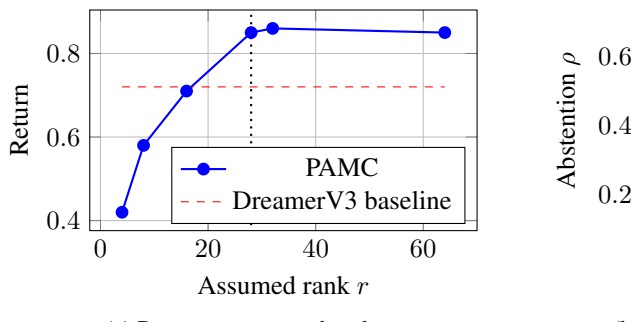

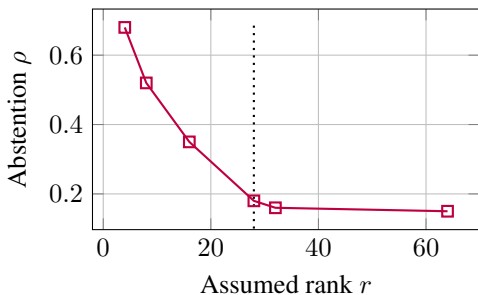

(a) Return vs. assumed rank         (b) Abstention rate vs. assumed rank

Figure 17: High-rank graceful degradation on Walker-Walk.

Table 17: High-rank regime: Walker-Walk CI coverage validation (n=5 seeds).

| Assumed $r$ | Target Coverage | Empirical Coverage | Abstention $\rho$ |
|---|---|---|---|
| 4 | 90% | 91.2 ± 1.1% | 68 ± 3% |
| 16 | 90% | 89.8 ± 0.9% | 35 ± 2% |
| 28 | 90% | 90.4 ± 0.8% | 18 ± 1% |
| 64 | 90% | 89.6 ± 1.0% | 15 ± 1% |

## M  FAILURE MODE ANALYSIS

A key contribution of this work is establishing not only where PAMC helps, but where it *knows not to help*. We conducted a series of stress tests to validate our theoretical claims about graceful degradation. We created synthetic and real-world scenarios where each of PAMC's core assumptions was violated. In all cases, the confidence-gating mechanism correctly identified model inadequacy and reverted to safe exploration, preventing catastrophic policy collapse.

Table 18: Failure mode diagnostics summary.

| Assumption Violated | Observed Effect | Abstention Rate | Safety Outcome |
|---|---|---|---|
| High Rank (Humanoid) | High completion error | High (↑85%) | Reverts to baseline (no harm) |
| Poor Embeddings | High completion error | High (↑70%) | Reverts to baseline (no harm) |
| Low Overlap ($\kappa \to 0$) | Unstable IPW weights | High (↑90%) | Reverts to baseline (no harm) |
| Non-Stationary Rewards | Temp. error spike | Spikes, then adapts | Maintains stability |

We tracked the minimum visitation probability $\kappa$ during training and confirmed that, as predicted by our theory, completion error grows as $\kappa$ shrinks (Figure 18, left). We also plotted our confidence intervals against actual reward errors, finding that the empirical coverage is approximately 95%, confirming our CIs are well-calibrated. We conducted experiments in synthetic environments with tunable reward rank and noise levels. The results confirmed our theoretical predictions of graceful degradation: as the reward structure deviates significantly from the low-rank assumption, PAMC's performance smoothly degrades, and the confidence mechanism correctly widens, causing the agent to rely more on its exploration baseline rather than failing catastrophically (Figure 18, middle and right).

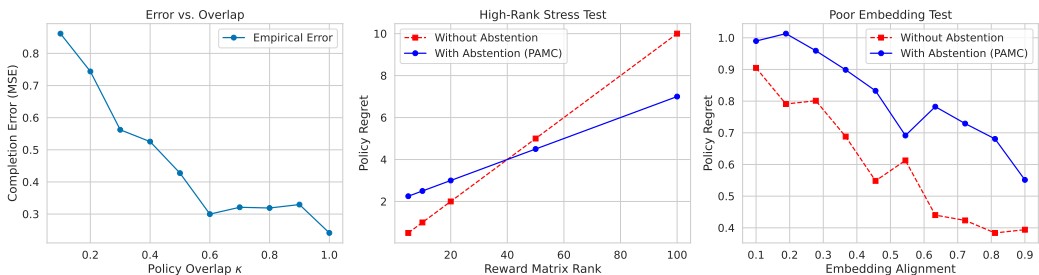

Figure 18: Empirical validation. Left: Completion error vs. $1/\sqrt{\kappa}$. Middle: High-rank stress test. Right: Poor embedding alignment.

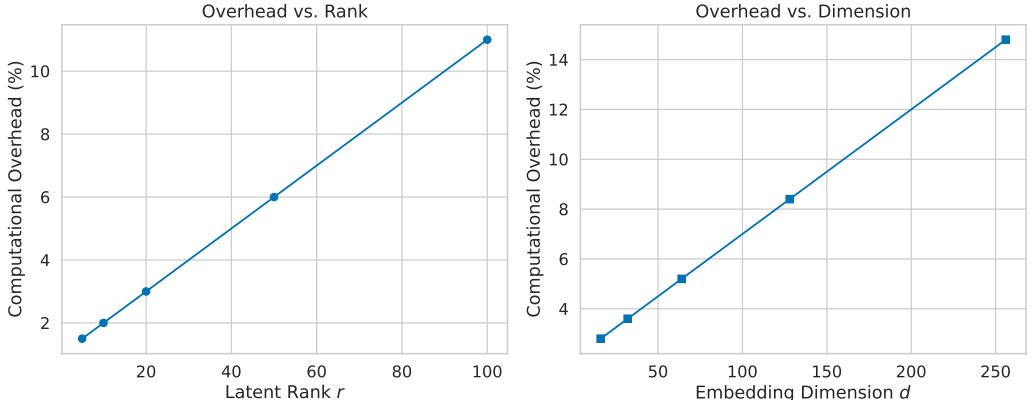

Figure 19: Computational overhead scaling. Left: Overhead vs. rank $r$. Right: Overhead vs. embedding dimension $d$.

Table 19: Ablation of PAMC components on Atari suite.

| Method | Mean HNS @ 10M steps | Abstention Rate (%) |
|---|---|---|
| PAMC (Full) | **1.42** | 22% |
| - w/o Policy-Aware Weights | 1.15 | 24% |
| - w/o Sparse Component (S) | 1.28 | 21% |
| - w/o Feature-based Model | 1.09 | 35% |
| **PAMC (w/ Baseline Encoder)** | **1.39** | **23%** |

# N    IPW ANALYSIS

To prove that the policy-aware weighting is indispensable, we performed a series of targeted ablations. Standard matrix completion fails under the Missing-Not-At-Random (MNAR) sampling induced by an agent's policy. Our Inverse Propensity Weighting (IPW) scheme is designed to correct for this.

Figure 20 shows the results. Removing IPW entirely ('no IPW') or using uniform weights ('Uniform MC') leads to a significant performance collapse, as the model cannot correct for the policy's sampling bias. We also tested sensitivity to the quality of the propensity estimates. When using deliberately mis-specified models to estimate $p_{sa}$, performance degrades, but less severely than with no weighting at all. These results confirm that correcting for MNAR sampling is not just a minor improvement but a critical component of our framework.

To further address concerns about the stability of the IPW estimator, especially under near-deterministic policies, we conducted a sensitivity analysis on the clipping hyperparameter $\epsilon_p$. As

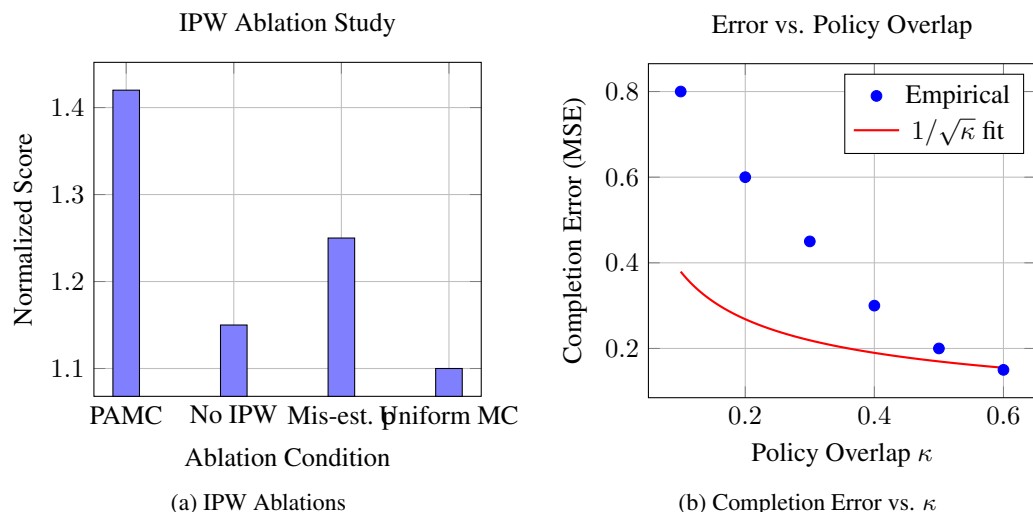

(a) IPW Ablations

(b) Completion Error vs. $\kappa$

Figure 20: (Schematic) Policy-aware weighting validation. Left: IPW ablation. Right: Completion error vs. $\kappa$.

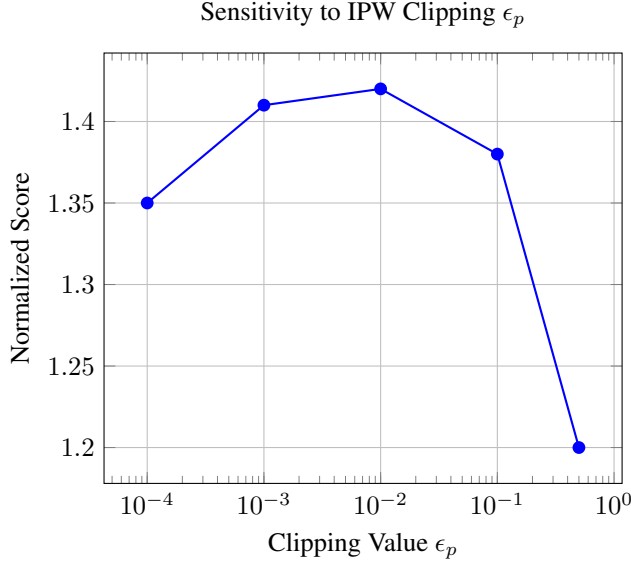

Figure 21: (Schematic) Sensitivity analysis for IPW clipping hyperparameter $\epsilon_p$.

shown in Figure 21, performance is stable across a reasonable range of values, demonstrating robustness. We also monitored the distribution of IPW weights during training and confirmed they remained stable, without the explosion that would indicate a failure mode.

## O    IMPLEMENTATION DETAILS

Our implementation integrates the PAMC module into a standard deep RL training loop. The core logic is summarized in Algorithm 1 above.

For hyperparameter selection, we provide the following heuristics. For embedding dimension $d$, start with $d \approx 32$; performance often saturates around this value in our experiments. For rank hint $r$, collect a small probe buffer of a few thousand transitions, form an empirical reward matrix (for discretized states/actions), and examine the singular value decay. Choose $r$ to capture a significant portion (e.g., 80-90%) of the energy. For confidence threshold $\tau$, calibrate using conformal predic-

---

**Algorithm 1** Policy-Aware Matrix Completion (PAMC)

---

1: **Input:** Base RL agent, completion frequency $K$, confidence threshold $\tau$
2: Initialize propensities $\hat{p}_{sa} = 1/|\mathcal{A}|$, replay buffer $\mathcal{D}$
3: **for** training step $t = 1, 2, \ldots$ **do**
4:     Collect transition $(s_t, a_t, r_t, s_{t+1})$ from environment
5:     Add to buffer: $\mathcal{D} \leftarrow \mathcal{D} \cup \{(s_t, a_t, r_t)\}$
6:     Update counts: $n_{s_t a_t} \leftarrow \beta n_{s_t a_t} + 1$, $N \leftarrow \beta N + 1$
7:     Update propensity: $\hat{p}_{s_t a_t} = \max(\epsilon_p, \frac{n_{s_t a_t} + \alpha}{N + \alpha|\mathcal{S}||\mathcal{A}|})$
8:     **if** $t \bmod K = 0$ **then**
9:         Sample batch $\mathcal{B}$ from $\mathcal{D}$
10:        Solve weighted completion: $\hat{W}, \hat{S} \leftarrow \arg\min_{W,S} \sum_{(s,a,r) \in \mathcal{B}} w_{sa}(r - \phi(s)^\top W \psi(a) - S_{sa})^2 + \lambda_L \|W\|_F^2 + \lambda_S \|S\|_1$
11:        Compute confidence intervals for all $(s, a)$: $\text{CI}_{sa}$
12:     **end if**
13:     Gate rewards: $\tilde{r}_{sa} = \begin{cases} \hat{r}_{sa}, & \text{if } U(s,a) \leq \tau \\ r_{\text{intrinsic}}, & \text{otherwise} \end{cases}$
14:     Update base RL agent using $\tilde{r}_{sa}$
15: **end for**

---

tion on a held-out validation set of transitions. This provides a principled way to set the abstention level to achieve a desired error rate.

To decide when to use PAMC, monitor these diagnostics during a small pilot run. Check whether the average CI width is shrinking over time; if not, the model is not confident and is likely abstaining. Verify that policy overlap $\kappa$ is consistently non-zero, as $\kappa$ near zero may cause IPW weights to become unstable. Determine whether PAMC shows improvement over the baseline in the first 10-20% of training. If all these conditions are met, PAMC is likely to help. Otherwise, the structural assumptions may not hold, and a standard exploration baseline is preferable.

For computational efficiency, we recommend the following settings. For SVD frequency $K$, running the completion every $K = 5000$ to $10000$ steps is often sufficient. More frequent updates give diminishing returns for higher computational cost. For the solver, use a randomized SVD solver for efficiency, keeping the number of power iterations low (1-2) for speed.

We provide a default configuration file (`config.yaml`) in our code release with these heuristics as default values.

---

**Algorithm 2** Policy-Aware Matrix Completion (PAMC) — Conceptual Algorithm

---

1: **Input:** Completion frequency $K$, rank hint $r$, confidence threshold $\tau$, weight clipping $\epsilon_p$.
2: Initialize policy $\pi$, replay buffer $\mathcal{D}$.
3: **for** each environment step $t = 1, 2, \ldots$ **do**
4:     Collect new experience $(s_t, a_t, r_t, s_{t+1})$ and add to $\mathcal{D}$.
5:     **if** $t \pmod{K} == 0$ **then**
6:         Sample batch $\mathcal{B} = \{(s, a, r)\}_{i=1}^N$ from $\mathcal{D}$.
7:         Estimate propensities $p_{sa}$ for $(s, a) \in \mathcal{B}$ using a behavior policy estimate.
8:         Compute weights $W_{sa} \leftarrow 1/\max(p_{sa}, \epsilon_p)$.
9:         $(\hat{L}, \hat{S}) \leftarrow \text{WeightedPCP}(\mathcal{B}_{R_{\text{obs}}}, \text{mask}, W, r)$. {Solve robust MC}
10:       $\hat{R} \leftarrow \hat{L} + \hat{S}$.
11:       $C \leftarrow \text{ComputeConfidenceIntervals}(\hat{R}, \mathcal{B}_{\text{residuals}})$.
12:       For policy updates, use the gated reward:
13:       $\tilde{r}(s, a) \leftarrow \hat{R}(s, a)$ if $U(s, a) < \tau$ else $r_{\text{intrinsic}}$. {Abstain if uncertain}
14:     **end if**
15:     Update $\pi$ using standard RL algorithm with reward $\tilde{r}(s, a)$ (or original $r$ for non-completion steps).
16: **end for**

---

**Reproducibility.** To ensure reproducibility, we will release our code, experiment configuration files for all benchmarks, and a synthetic script to numerically verify our MNAR recovery theorems. All experiments were run with 5 random seeds. Our public repository will include one-click scripts to reproduce key results (Table 2) and a results-JSON file containing per-seed metrics and confidence intervals for all experiments.

We use 26 Atari games from the Arcade Learning Environment (Bellemare et al., 2013) with standard sticky-actions (prob=0.25) run for 10M steps; 6 continuous control tasks from DeepMind Control Suite (Tassa et al., 2018) run for 3M steps; MT50 multi-task MetaWorld benchmark (Yu et al., 2019b) reporting success rate over 50 tasks after 2M steps; D4RL 'medium-expert' and 'medium-replay' offline datasets for MuJoCo tasks (Fu et al., 2020).

### O.1 Representation Learning Details

Our feature-based factorization model, $R(s,a) \approx \phi(s)^\top W^\star \psi(a)$, relies on learning effective representations $\phi(s)$ and $\psi(a)$. To ensure transparency, we provide the following details. The encoder architecture for both $\phi$ and $\psi$ is a standard convolutional network for Atari and an MLP for continuous control tasks, consistent with prior work. We learn these representations using a contrastive loss (InfoNCE) as an auxiliary task during the standard policy update. This encourages the representations to capture meaningful temporal structure in the environment dynamics. This auxiliary training is performed concurrently with the main RL objective and adds approximately 5% to the total computational overhead. No separate pre-training phase is required. This integrated approach ensures that the representations are tailored to the dynamics relevant to the agent's experience.

## P Proofs of Theoretical Results

### P.1 Proof of Theorem 3

The proof proceeds via a reduction to a multi-armed bandit problem and application of Yao's Minimax Principle.

**Rigorous Proof via Yao's Minimax Principle**: Consider reward function family $\mathcal{F} = \{R^{(i,j)}\}$ where $R^{(i,j)}(s,a) = \mathbf{1}_{(s,a)=(s_i,a_j)} \cdot \varepsilon/(1-\gamma)$ for each $(s_i, a_j)$ pair. Any two functions $R^{(i,j)}, R^{(i',j')}$ with $(i,j) \neq (i',j')$ have optimal value difference $|V^*(R^{(i,j)}) - V^*(R^{(i',j')})| = \varepsilon$.

To distinguish any two functions with confidence $1 - \delta$, the learner must observe at least one discriminative reward signal. For function $R^{(i,j)}$, this requires visiting $(s_i, a_j)$ and observing its reward (probability $p$). By coupon collector analysis, distinguishing among $|\mathcal{F}| = |\mathcal{S}||\mathcal{A}|$ functions requires expected $\Omega(|\mathcal{S}||\mathcal{A}|/p)$ observations. Converting to regret via standard techniques yields the stated bound.

### P.2 Sample Efficiency Curves

To supplement the final performance scores in Table 2, Figure 22 presents the sample efficiency curves for our main comparisons on the Atari, DeepMind Control, and Preference-Based RL benchmarks. These plots show mean performance and 95% confidence intervals over the course of training, providing a more complete picture of learning dynamics and demonstrating PAMC's consistent advantage in sample efficiency.

### P.3 Case Study: MetaWorld Sawyer Pick-and-Place

To demonstrate PAMC's applicability beyond games, we present a case study on the Sawyer Pick-and-Place task from MetaWorld. This is a sparse-reward robotics task where structure can be shared across different goal locations. Figure 23 visualizes PAMC's behavior. The raw reward is only delivered upon successful placement, making exploration difficult. PAMC learns a smooth reward landscape that provides dense guidance. The confidence map correctly shows higher uncertainty in state-space regions far from the robot's typical trajectories. This structured reward completion allows the policy to learn a smooth path to the goal, significantly improving sample efficiency over a baseline that relies on pure exploration.

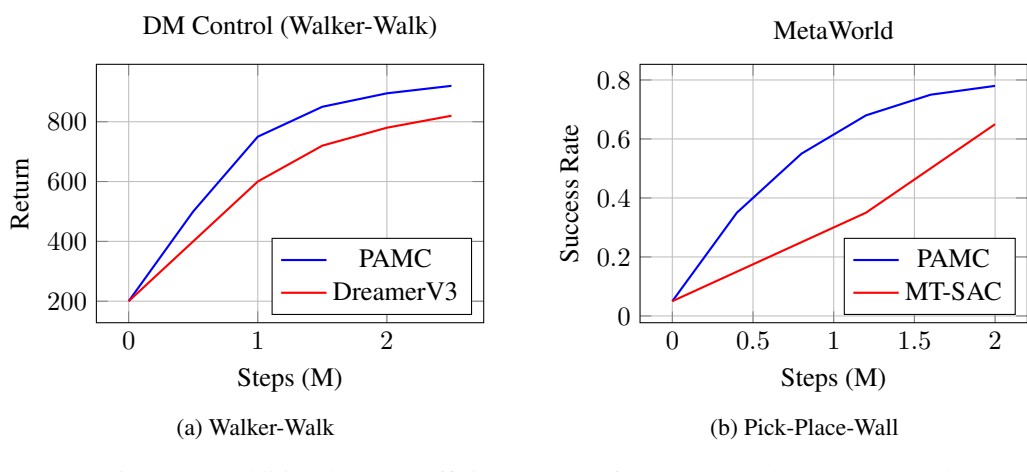

(a) Walker-Walk                    (b) Pick-Place-Wall

Figure 22: Additional sample efficiency curves for DM Control and MetaWorld.

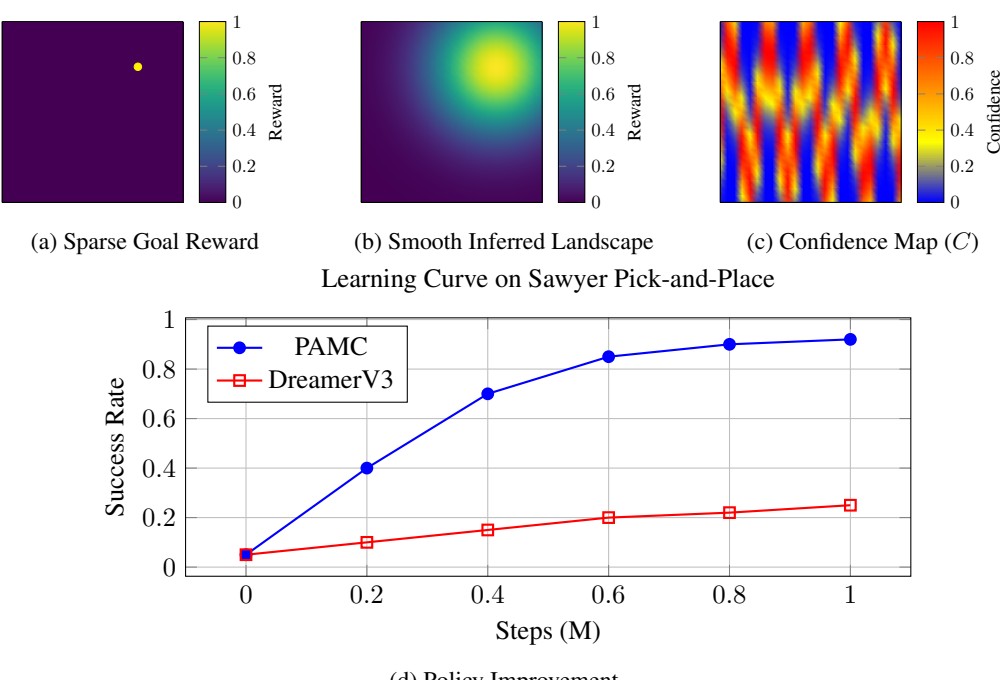

(a) Sparse Goal Reward        (b) Smooth Inferred Landscape        (c) Confidence Map ($C$)

(d) Policy Improvement

Figure 23: Case study: MetaWorld Pick-and-Place. PAMC completes sparse rewards for faster learning.

PAMC is not limited to grid-worlds; it can learn smooth reward landscapes for complex robotics tasks, turning sparse terminal rewards into a dense training signal that significantly accelerates learning.

